# *Clostridium butyricum* RH2 ameliorates diarrhea in juvenile mice under continuous antibiotic exposure by modulating gut microbiota and metabolome

Yufeng Zhao,[1] Kan Gao,[1] Yunlong Shang,[1] Shen Cheng,[1] Qianlei Ren,[1] Fengyu Guo,[1] Yu Wang[1]

**ABSTRACT** Antibiotic-associated diarrhea (AAD) is a self-limiting disorder triggered by antibiotic therapy in pediatric populations. Although multiple probiotics are clinically employed for AAD management, the therapeutic efficacy of *Clostridium butyricum* (*C. butyricum*) in pediatric AAD and its underlying mechanisms remain poorly characterized. This study aimed to establish a juvenile mice model of AAD and investigate the therapeutic potential of oral *C. butyricum* administration in juvenile mice subjected to continuous antibiotics exposure. We systematically assessed pathological changes in colonic tissue, colitis severity, intestinal epithelial barrier integrity, fecal metabolomic profiles, and gut microbiota diversity. Our analysis demonstrates that *C. butyricum* ameliorates intestinal inflammation, enhances barrier function by modulating the gut microbiota and its metabolites, and significantly alleviates diarrhea symptoms in juvenile AAD mice. Collectively, these findings indicate that the therapeutic benefits of *C. butyricum* are closely linked to its ability to tolerate continuous antibiotic exposure, providing a scientific rationale for its co-administration with antibiotics.

**IMPORTANCE** *Clostridium butyricum* demonstrates significant therapeutic potential for pediatric antibiotic-associated diarrhea (AAD) by dual modulation of gut microbiota and host physiology. This study reveals its capacity to alleviate intestinal inflammation, restore barrier integrity via upregulation of tight junction proteins and mucins, and rebalance gut microbiota linked to key anti-inflammatory metabolites—even under ongoing antibiotic exposure. These findings position *C. butyricum* as a targeted probiotic therapy for AAD, offering mechanistic insights to advance microbiome-driven interventions for antibiotic-induced diarrhea in children.

**KEYWORDS** antibiotic-associated diarrhea, gut barrier function, *Clostridium butyricum* RH2, gut microbiota, metabolomics

A s the most frequent complication of antimicrobial treatment, antibiotic-associated diarrhea (AAD) is defined as unexplained diarrheal episodes temporally linked to antibiotic use. All antibiotic classes demonstrate diarrheagenic potential, particularly broad-spectrum agents, including penicillins, cephalosporins, and clindamycin (1). The clinical manifestations depend on the encompassing antibiotic class, dosage regimen, treatment duration, administration route, and combination therapies (2). While AAD affects all age demographics, pediatric populations exhibit heightened susceptibility because their immature intestinal ecosystem is more vulnerable to microbial perturbations. Recent studies have also highlighted that antibiotic-induced short-chain fatty acid (SCFA) depletion, bile acid dysregulation, and epithelial barrier disruption are key contributors to pediatric AAD (3). Epidemiological surveillance data indicate substantial variability across care settings: US pediatric cohorts demonstrate 6% incidence

**Peer Reviewer** Sahar M. Jawad Abduladheem, University of Kufa, Al Najaf, Iraq

Address correspondence to Yu Wang, wangyu@hzydsw.cn.

Y.Z., K.G., Y.S., S.C., F.G., Q.R., and Y.W. are employees of Hangzhou Grand Biologic Pharmaceutical Inc. Although the investigational product was sourced from the authors' affiliated institution, all experimental procedures strictly adhered to Good Laboratory Practice (GLP) guidelines, ensuring methodological rigor and data impartiality. Institutional quality assurance protocols were implemented throughout the study to maintain research integrity independent of corporate affiliations.

in ambulatory care versus 80% in hospitalized patients (4). Chinese clinical studies, currently limited to inpatient settings, report incidence rates ranging from 16.8% to 70.6% (5). Current pediatric guidelines primarily emphasize the discontinuation of causative antibiotics whenever clinically feasible (6). However, discontinuing antibiotics is often not feasible in cases of severe infection or when prophylactic medication is required. Therefore, identifying adjunctive interventions capable of alleviating AAD under ongoing antibiotic exposure is clinically important.

Probiotics are defined as live microorganisms that confer health benefits to the host when administered in adequate doses (7). Substantial evidence from preclinical and clinical studies has established that antibiotic-induced gut microbiota dysbiosis plays a fundamental role in AAD pathogenesis. This mechanistic understanding supports probiotics as viable therapeutic agents for microbiota restoration. Multiple randomized controlled trials have demonstrated the efficacy of specific microbial strains in AAD prevention, including *Bifidobacterium* spp., *Lactobacillus* spp., and the fungal species *Saccharomyces boulardii* (8–13).

*Clostridium butyricum* (*C. butyricum*) is a strictly anaerobic, spore-forming bacterium that has been widely used as a probiotic in Asia for decades. It is commonly found in soil, dairy products, plant matter, and the human gut, where it colonizes early in life and is present in 10%–20% of adults (14). Different strains of *C. butyricum* confer various health benefits through interactions with the host (15), and growing evidence supports its role as a multifunctional therapeutic agent with potential applications in gastrointestinal, neurological, metabolic, and even anticancer treatments (16). Among its known functions, production of butyrate and other SCFAs suggests possible links to epithelial barrier protection and immune modulation, although strain-specific effects remain insufficiently understood. It should be noted, however, that unlike widely recognized generally recognized as safe (GRAS) genera, such as *Bifidobacterium* and *Lactobacillus*, *Clostridium* species have not been universally granted GRAS status for probiotic applications. Nevertheless, specific strains (such as MIYAIRI 588 strain, CGMC0313.1 strain, etc.) have been shown to be safe in humans based on extensive clinical use and historical data (17–20). Despite these findings, the therapeutic effects of the RH2 strain on AAD, particularly under continuous antibiotic pressure, have not been characterized, and its metabolic or barrier-related pathways in this context remain unclear.

In this study, we employed *C. butyricum* RH2 strain to investigate its therapeutic efficacy and underlying mechanisms in a juvenile murine model of AAD. Given that pediatric populations are particularly vulnerable to AAD complications, we modeled this using 4-week-old C57BL/6J mice at a comparable developmental stage. The model was induced through sequential oral administration of an antibiotic cocktail (ampicillin, streptomycin, and clindamycin), simulating pediatric clinical conditions. By integrating microbiota profiling, epithelial barrier markers, and metabolomics, this study aimed to evaluate both the therapeutic potential of RH2 and the possible microbiota–metabolite–barrier relationships that may contribute to its effects. Our findings demonstrate that *C. butyricum* RH2 exhibits significant potential as a probiotic adjuvant for maintaining intestinal homeostasis in antibiotic-dependent pediatric AAD management.

## MATERIALS AND METHODS

### Reagents and probiotics strain

The juvenile AAD murine model was established through daily gavage administration of an antibiotic cocktail containing clindamycin (Jiudian, Hunan, China), ampicillin (United Laboratories, Hong Kong, China), and streptomycin (Lukang, Shandong, China) dissolved in normal saline. Antibiotic cocktail administration was performed daily at 10:00.

*C. butyricum* RH2 lyophilized powder (Grand Biologic Pharmaceutical; Batch J202304072) was reconstituted in sterile 0.9% NaCl solution to achieve target concentrations. A minimum 4-h interval was maintained between probiotic and antibiotic administrations to prevent pharmacological interference.

## Animal experimental design

Sixty specific pathogen-free male C57BL/6J mice (3-week-old) were obtained from Shanghai SLAC Laboratory Animal Co., Ltd. (Shanghai, China). The animals were maintained under controlled environmental conditions (temperature: 22°C ± 2°C; humidity: 40% ± 5%) with a 12-h light/dark cycle and provided *ad libitum* access to food and water.

As illustrated in Fig. 1A, following a 7-day acclimatization period, mice were randomly allocated into six experimental groups: normal control (NC), antibiotic-associated diarrhea model (M), antibiotic withdrawal control (AW), and three *C. butyricum* RH2 treatment groups receiving low (LD: $1.05 \times 10^5$ CFU daily per mice), medium (MD: $1.05 \times 10^7$ CFU daily per mice), or high doses (HD: $1.05 \times 10^9$ CFU daily per mice). Based on established protocols demonstrating successful AAD model induction through 3-day antibiotic administration (21), therapeutic interventions commenced post-antibiotic treatment immediately and continued for 14 days. To prevent spontaneous recovery reported in untreated AAD models (21), antibiotic administration was maintained throughout the treatment phase. The AW group served as a self-recovery control, receiving antibiotics for 3 days, followed by 14 days of saline treatment. Fecal samples were collected from all groups for subsequent analyses.

## Diarrhea assessment

Diarrhea severity was quantitatively assessed through fecal consistency scoring and fecal moisture content analysis following established methodology (22). Antibiotic-associated diarrhea status was classified based on the following visual grading scale: (i) alert behavioral status with formed, the feces are elliptical in shape, with a hard texture and a brownish color, score = 1; (ii) general mental state, the feces are sausage-shaped, with a smooth surface and a yellowish color, score = 2; and (iii) bad mental state, the feces are in the shape of sausages or paste, with a wet and soft texture, and a yellowish color, score =3. Fresh fecal samples were collected over a standardized 1-h period using metabolic cages. We immediately weighed the samples to obtain the wet weight, desiccated them at 60°C until they reached a constant mass, and then reweighed them to obtain the dry weight. Fecal moisture content was calculated using

Water in Stool (%) = [(Wet Weight − Dry Weight)/Wet Weight] × 100.

## RNA extraction and RT-PCR

Total RNA was isolated from tissue specimens using TRIzol Reagent (Thermo Fisher Scientific, Waltham, MA, USA) following the manufacturer's protocol. The concentration and purity of the extracted RNA were determined using a spectrophotometer. First-strand cDNA synthesis was synthesized from 0.5 µg of total RNA using the HiScript III Reverse Transcriptase Kit (Vazyme Biotech Co., Ltd., Nanjing, China). Quantitative real-time polymerase chain reaction (qPCR) was performed in a 10 µL reaction volume containing 1 µL of cDNA template, 5 µL of ChamQ Universal SYBR qPCR Master Mix (Vazyme Biotech Co., Ltd.), 0.4 µL of each primer (10 µM), and 3.2 µL of nuclease-free water. The reactions were run on a QuantStudio 7500 Fast Real-Time PCR System (Applied Biosystems, Foster City, CA, USA) under the following thermal cycling conditions: initial denaturation at 95°C for 30 s, followed by 40 cycles of 95°C for 10 s and 60°C for 30 s. A melt curve analysis was subsequently performed to confirm amplification specificity. Gene expression levels were normalized to GAPDH and calculated using the $2^{(-\Delta\Delta Ct)}$ method, as previously described (23). The primer sequences are shown in Table S1.

## Histological examination

For histological analysis, colon tissue samples were fixed in 4% paraformaldehyde and embedded in paraffin. Hematoxylin and eosin (H&E) staining was performed according to established protocols (23).

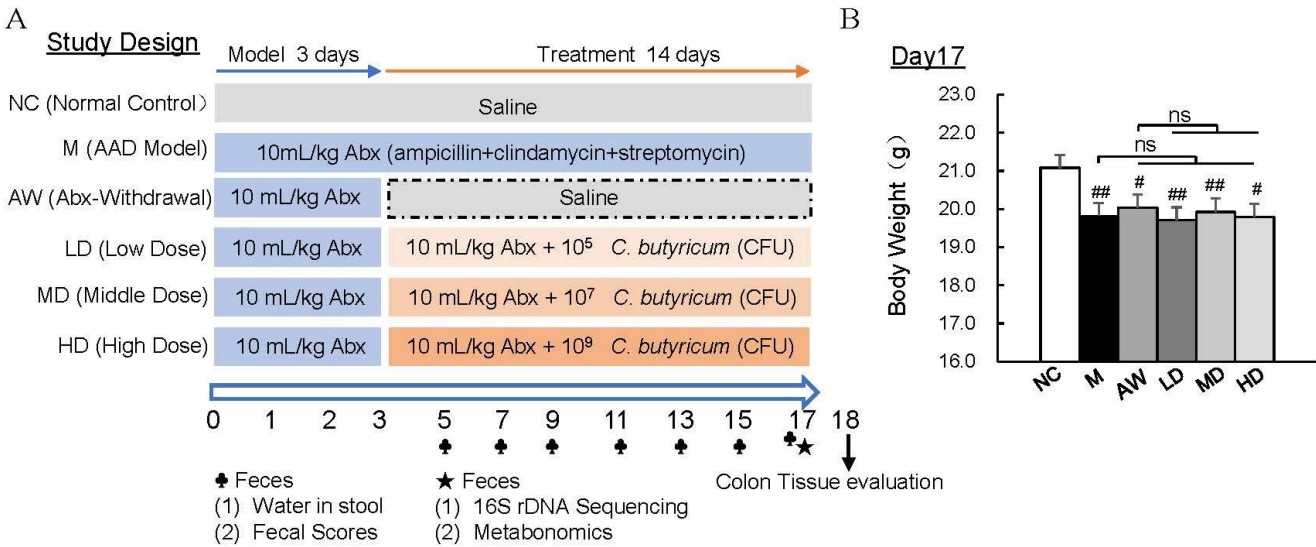

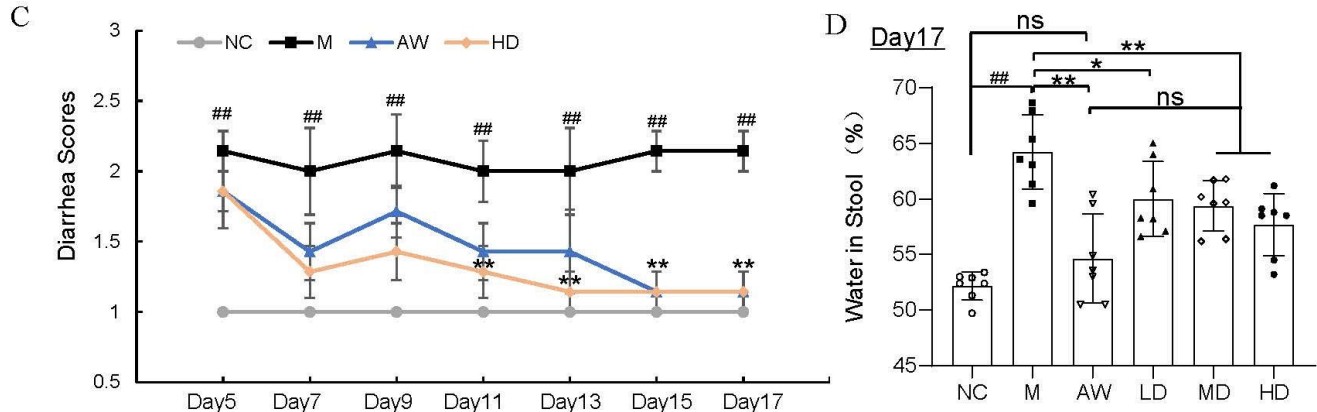

FIG 1 *C. butyricum* RH2 ameliorated AAD symptoms in juvenile mice. (A) Schematic diagram in the animal experiment. (B) Body weight. Data were presented as the means ± SEM, #$P < 0.05$, ##$P < 0.01$ vs NC; ns, no significance. (C) Diarrhea score for the duration of treatment. Data were presented as the means ± SEM, ##$P < 0.01$ vs NC; **$P < 0.01$ vs M. (D) Water in mice stool. Data were presented as the means ± SD, ##$P < 0.01$ vs NC; *$P < 0.05$, **$P < 0.01$ vs M; ns, no significance.

## 16S rRNA gene amplicon sequencing

Extract bacterial DNA from mice feces was obtained using OMEGA Soil DNA Kit (M5635-02; OMEGA Bio-Tek, Norcross, GA, USA). NanoDrop NC2000 spectrophotometer (Thermo Fisher Scientific, Waltham, MA, USA), followed by agarose gel electrophoresis, was used to measure the quality of extracted DNA. PCR amplification was performed on the V3–V4 region of the bacterial 16S rRNA gene using forward primer 338F (5′-AC TCCTACGGGAGGCAGCA-3′) and reverse primer 806R (5′-GGACTACHVGGGTWTCTAAT-3′). Vazyme VAHTSTM DNA Clean Beads (Vazyme, Nanjing, China) were used to purify PCR amplicons, which were quantified with QuantiT PicoGreen dsDNA assay kit (Invitrogen, Carlsbad, CA, USA). Purified amplicons were merged in equal quantities and paired for sequencing using Illumina NovaSeq platform and NovaSeq6000 SP kit (500 cycles). The sequence data were processed using QIIME2 according to previously described methods (24).

## Metabolomics analysis

Metabolites in fecal samples were quantified using an ultra-performance liq-uid chromatography-tandem mass spectrometry (UPLC-MS/MS) system (ACQUITY

UPLC-Xevo TQ-S; Waters Corporation, Milford, MA, USA) with the Q300 Metabolite Array Kit (Metabo-Profile Biotechnology, Shanghai, China). Raw data files generated by UPLC-MS/MS were processed using the iMAP platform (v1.0; Metabo-Profile, China). Multivariate statistical analysis (ANOVA or Kruskal-Wallis test, selected based on data normality and variance homogeneity) was applied to perform principal component analysis (PCA) and identify differential metabolites between groups. Differential metabolites were defined by a significance threshold of $P < 0.05$. Shared metabolites among pairwise comparisons (NC vs M, M vs AW, and M vs HD) were visualized using Venn diagrams, while hierarchical clustering analysis illustrated metabolite expression patterns across groups. Metabolite enrichment analysis for *C. butyricum* RH2-reversed metabolic alterations was conducted through the MetaboAnalyst platform.

## Statistics

All data were expressed as the mean ± standard error of the mean (SEM). One-way ANOVA among groups, followed by Student's *t*-test between groups, was performed to analyze the differences using the Prism 9.0 program (GraphPad Software, San Diego, Canada). Spearman correlation analyses between gut microbiota composition, fecal metabolites, and AAD indices were performed using SPSS version 24 (IBM Corp., Armonk, NY, USA). Statistical significance was defined as $P < 0.05$.

## RESULTS

### *C. butyricum* RH2 ameliorated antibiotic-associated diarrhea symptoms in juvenile mice

To investigate the therapeutic effects of *C. butyricum* RH2 on juvenile AAD, we established a murine model through 3-day oral administration of triple-antibiotic cocktails. Subsequently, different doses of *C. butyricum* RH2 were administered orally for 14 days without antibiotic discontinuation. A positive control group receiving a 14-day antibiotic withdrawal post-modeling was included. The experimental design is schematically presented in Fig. 1A. After establishing the AAD model and assigning mice into the respective groups, we evaluated the general health status of the animals. AAD juvenile mice exhibited significant weight loss compared to healthy controls, accompanied by clinical manifestations, including lethargy and reduced activity. Neither *C. butyricum* RH2 supplementation nor antibiotic withdrawal reversed this weight deficit (Fig. 1B). Fecal samples collected every 48 h during treatment were evaluated using standardized diarrhea indices: fecal consistency scores and water in stool. The model group maintained elevated fecal scores throughout the intervention, whereas antibiotic withdrawal and high-dose *C. butyricum* RH2 groups demonstrated progressive score reduction (Fig. 1C). Moreover, fecal scores also showed that supplementing different doses of *C. butyricum* RH2 can alleviate diarrhea, but there is no dose-dependent effect (Fig. S1A). Consistently, the water in the stool of AAD juvenile mice significantly increased compared to the normal group, and different treatment groups significantly reduced the water in the stool of AAD juvenile mice after the therapy (Fig. 1D). More importantly, supplementing *C. butyricum* RH2 can significantly improve the symptoms of pathological AAD in juvenile mice in a dose-dependent manner from the indicator of water in stool (Fig. 1D).

### *C. butyricum* RH2 mitigated intestinal inflammation of colon tissues

Colon tissues were dissected from juvenile mice and measured for length analysis. Compared with normal controls, AAD mice exhibited significant colon shortening. Both antibiotic withdrawal and *C. butyricum* RH2 supplementation effectively restored colon length to normal dimensions (Fig. 2A and B). Given the established correlation between colon shortening and inflammatory processes, we subsequently performed histopathological examination through H&E staining. Figure 2C illustrates the characteristic

histopathological changes across experimental groups. Normal controls displayed intact mucosal architecture with regularly arranged epithelial cells (Fig. 2C, upper left panel). In contrast, AAD mice exhibited severe inflammatory infiltration and mucosal disruption (Fig. 2C, upper right panel). Both therapeutic interventions—antibiotic cessation (Fig. 2C, lower left panel) and *C. butyricum* RH2 administration (Fig. 2C, lower right panel)—substantially improved histological parameters, demonstrating comparable restorative effects. Complementary qPCR analysis confirmed the anti-inflammatory effects of *C. butyricum* RH2 through modulation of key cytokine expression. Treatment significantly normalized mRNA levels of pro-inflammatory mediators (*TNF-α, IL-1β, IL-6*) and anti-inflammatory regulators (*IL-4, IL-10*) in colonic tissues (Fig. 2D). These findings collectively demonstrate that therapeutic intervention through either antibiotic discontinuation or *C. butyricum* RH2 supplementation effectively attenuates inflammatory cell infiltration and mitigates intestinal inflammation in AAD mice.

## *C. butyricum* RH2 improves gut barrier integrity in AAD juvenile mice

Given the established association between intestinal inflammation and barrier dysfunction, we further investigated the therapeutic potential of *C. butyricum* RH2 in preserving gut barrier integrity. Downregulation of epithelial barrier biomarkers—including occludin, claudins, zonula occludens, mucin, and E-cadherin—serves as a hallmark of intestinal barrier impairment. qPCR was performed and verified that the mRNA expression of tight junctions (*Cldn1, Cldn5, Tjp1, and Ocln*), adherent junction (*Cdh1*), and mucin secretion (*Muc2*) markers in the colon were recovered by *C. butyricum* RH2, indicating functional restoration of barrier components (Fig. 3). Notably, *C. butyricum*

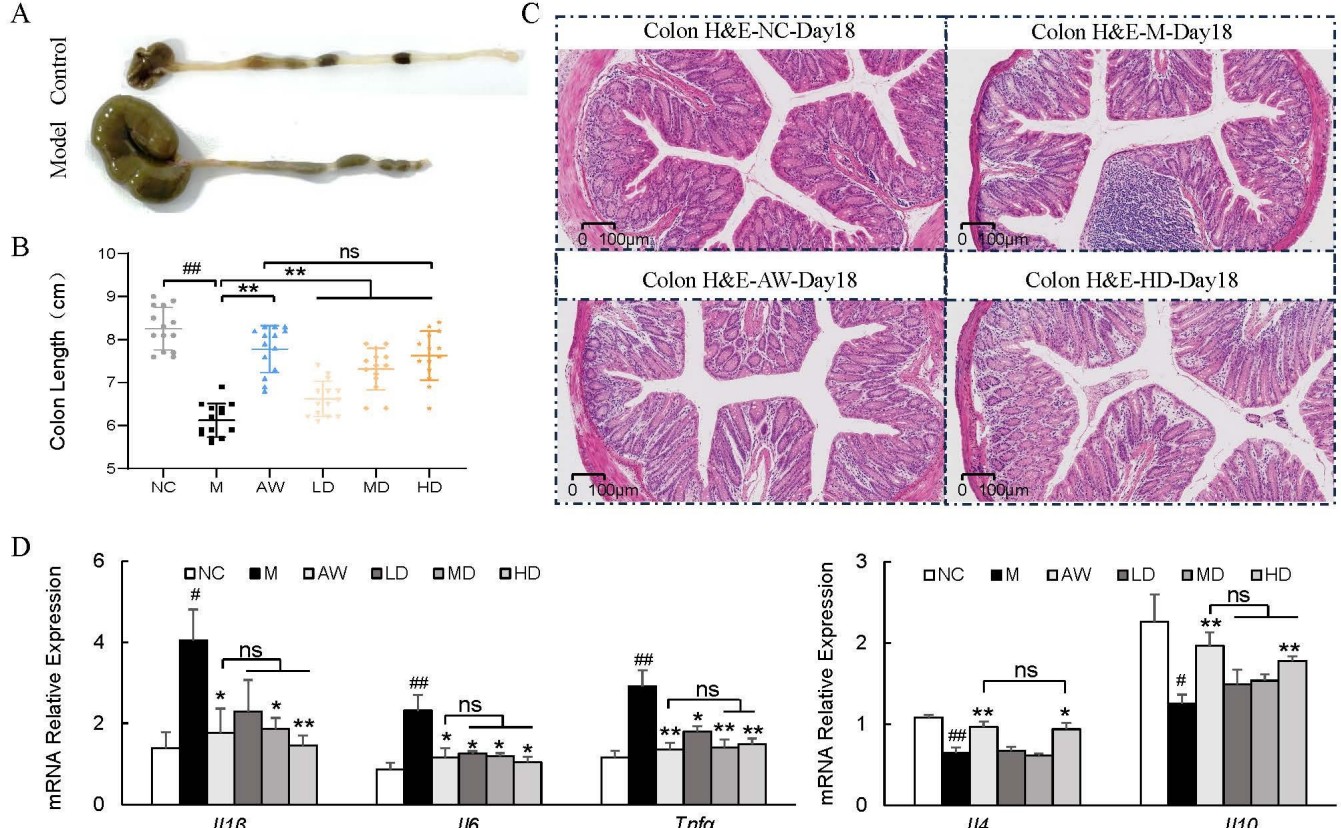

FIG 2 *C. butyricum* RH2 mitigated intestinal inflammation of colon tissues. (A) Representative pictures of colons and colon length. (B) Colon length (*n* = 14). Data were presented as the means ± SD, **P < 0.01 vs M; ##P < 0.01 vs NC; ns, no significance. (C) Colon picture of H&E staining. Scale bar: 100 µm. (D) Relative mRNA levels of inflammatory-related gene expression in the colon (*n* = 5–8). Data were presented as the means ± SEM, *P < 0.05, **P < 0.01 vs M; #P < 0.05, ##P < 0.01 vs NC; ns, no significance.

RH2 exhibited dose-dependent efficacy in barrier restoration, with high-dose supplementation demonstrating superior effects to low-dose treatment. Intriguingly, 14-day antibiotic withdrawal partially restored mRNA levels of these barrier-associated markers in AAD mice, yet failed to achieve full normalization (Fig. 3). This incomplete recovery of barrier-related gene expression highlights the long-term detrimental effects of antibiotic exposure.

## *C. butyricum* RH2 alters gut microbiota composition and functionality in juvenile AAD mice

To elucidate the microbial regulatory effects of *C. butyricum* RH2, we conducted 16S rRNA sequencing on fecal samples from juvenile AAD mice. Alpha diversity indices (observed species, Chao1, Faith's PD, and Good's coverage) were employed to assess microbial richness and community diversity. Although there was no significant change between the model group and all four treatment groups (Fig. S2), we found that compared to the normal group, the observed species, Chao1, and Faith's PD indices in the model group decreased, while the Good's coverage index in the model group increased (Fig. 4A). These data indicate that antibiotics could reduce the bacterial species richness and community diversity in stool of mice.

The principal coordinate analysis (PCoA) based on unweighted Uni-Frac distances (accounting for the abundance of OTUs) indicates that the β-diversity values can be used to clearly discriminate between normal mice and AAD mice. Antibiotic withdrawal, or high level of *C. butyricum* RH2 ($10^9$ CFU/day) treatment, resulted in further discrimination compared to AAD mice, with a β-diversity value that was close to that of normal mice (Fig. 4B). Histograms illustrating the gut microbial community structure revealed the microbial species and their relative abundance. *Bacteroidetes*, *Proteobacteria,* and *Firmicutes* were the dominant phyla in the six groups (Fig. 4C). The proportion of *Bacteroidetes* and *Firmicutes* was the highest in the control, antibiotic-withdraw group,

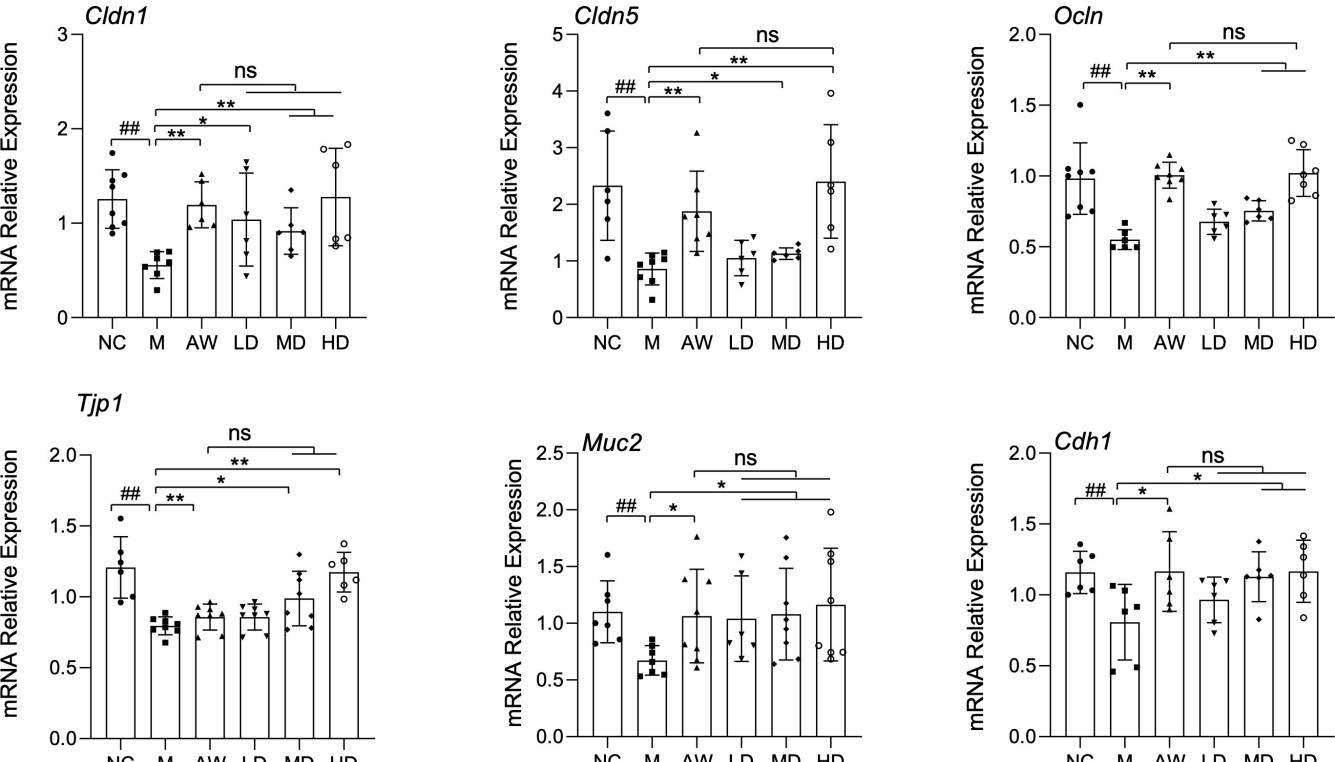

**FIG 3** *C. butyricum* RH2 improves gut barrier integrity in AAD juvenile mice. Relative mRNA levels of inflammatory-related gene expression in the colon ($n = 5–8$). Data were presented as the means ± SD, *$P < 0.05$, **$P < 0.01$ vs M; ##$P < 0.01$ vs NC; ns, no significance.

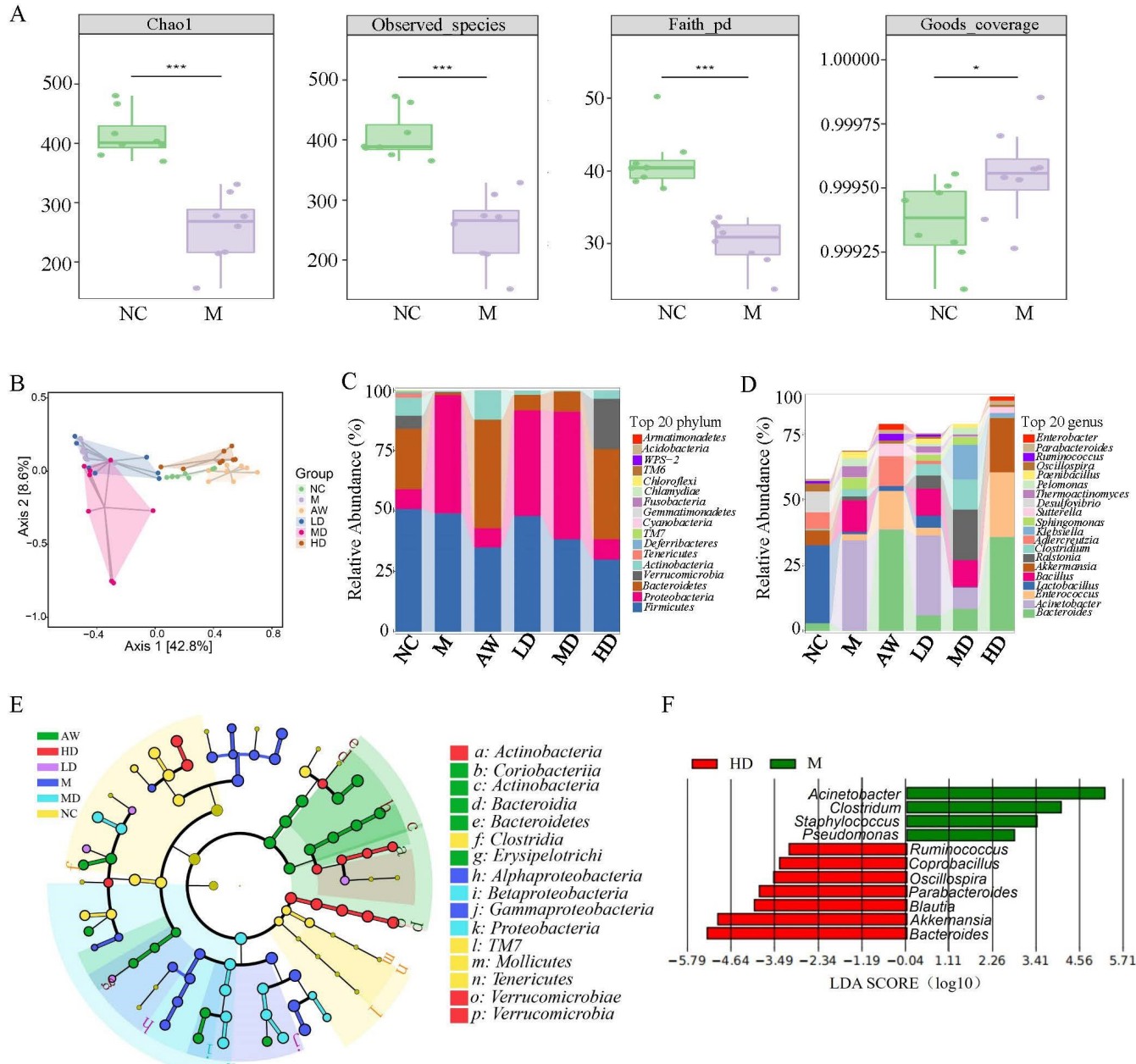

FIG 4 *C. butyricum* RH2 alters gut microbiota composition and functionality in juvenile AAD mice. (A) Alpha diversity of the gut microbiota. Kruskal-Wallis tests were performed to analyze the differences, ***$P < 0.001$ vs NC. (B) Principal coordinate analysis (PCoA) of unweighted UniFrac. Samples are colored by different treatments. (C) Relative abundance of phyla. (D) Relative abundance of genera. (E) Cladogram generated from linear discriminant analysis effect size (LEfSe) analysis. (F) LEfSe identifies differential abundance of bacteria between *C. butyricum RH2* treatment mice and AAD mice.

and high level of *C. butyricum* RH2 ($10^9$ CFU/day) treatment group, while *Proteobacteria* was the dominant phylum in the model group (Fig. 4C).

In addition, the dynamic alterations of fecal microbial composition at the genus level (top 20) were summarized in Fig. 4D. Compared to other groups, the model group had the highest abundance of *Acinetobacter* and *Bacillus* and the lowest abundance of *Bacteroides* (Fig. 4D). Supplementing *C. butyricum* RH2 helps to increase the number of *Bacteroidetes* in AAD mice and decrease the number of *Acinetobacter* and *Bacillus*, which is consistent with the effect of antibiotic-withdraw mice (Fig. 4D). LEfSe analysis, a method for identifying biomarkers that explain compositional differences in microbial

communities, revealed significant differences in the taxa found between juvenile AAD mice and normal mice, antibiotic withdrawal mice, and *C. butyricum* RH2 treated mice (Fig. 4E). Importantly, our findings unveiled that a significantly increased abundance of potential health-promoting microorganisms, such as species belonging to *Ruminococcus, Akkermansia, and Oscillospira,* in the $10^9$ CFU/day *C. butyricum* RH2 treatment group (Fig. 4F).

In short, these results demonstrate that *C. butyricum* RH2 administration ($10^9$ CFU/day) effectively counteracts antibiotic-driven microbiota disruption, achieving microbial restoration comparable to spontaneous recovery through antibiotic withdrawal.

## *C. butyricum* RH2 partially restored the alteration of fecal metabolites in juvenile AAD mice

Fecal metabolite profiling across six experimental groups was conducted using UPLC-MS. We identified 216 metabolites, with amino acids and fatty acids constituting the predominant chemical classes (Fig. S3). Stacked bar chart analysis revealed significant reductions in organic acids and SCFAs in the model group compared to other experimental groups (Fig. 5A). Both antibiotic withdrawal and *C. butyricum* RH2 supplementation demonstrated dose-dependent restoration of organic acid and SCFA levels in AAD mice (Fig. 5A).

Multivariate and univariate statistical analyses were performed to identify significantly changed metabolites among groups. Multivariate analysis through PCA revealed that there were distinct separations among the normal control mice, AAD mice, antibiotic withdrawal mice, and supplementing *C. butyricum* RH2 mice, which indicated that profiles of metabolites exhibited different patterns among the groups (Fig. 5B). The differential metabolites of pairwise groups (NC vs M, M vs AW, and M vs HD) obtained using univariate statistical analysis, and threshold value for differential metabolites selection is $P < 0.05$. A total of 130 overlapping metabolites were obtained between the three paired groups mentioned above (Fig. 5C). These 130 overlapping metabolites include fatty acids, amino acids, organic acids, carbohydrates, bile acids, and SCFAs (Fig. 5D).

Moreover, hierarchical clustering revealed that 130 metabolites exhibited opposite patterns among NC, M, AW, and HD groups (Fig. 5E), which means that high levels of *C. butyricum* RH2 ($10^9$ CFU/day) can significantly restore changes in fecal metabolites in AAD mice. Enriched metabolic pathways were mapped and analyzed via the SMPDB, which identified three significantly affected metabolic pathways: α-linolenic acid metabolism, linoleic acid metabolism, and urea cycle/aspartate metabolism (Fig. 5F).

## *C. butyricum* RH2-mediated restoration of gut microbiota-metabolite interactions associates with AAD symptom amelioration

Spearman's correlation analysis revealed significant associations between *C. butyricum* RH2-modulated gut microbiota, fecal metabolites, and clinical AAD parameters. The results revealed that most of the *C. butyricum*-restored gut microbiota and its metabolites were significantly correlated with AAD-related indexes. A majority of the fecal metabolites SCFAs, fatty acids, organic acids, carbohydrates, and amino acids, in particular of the butyric acid, rhamnose, linoelaidic acid, and palmitoleic acid, were strongly negatively correlated with water in stool, diarrhea score, proinflammatory factor, whereas they were positively correlated with tight junctional proteins, mucus proteins, and suppressible inflammatory factor (Fig. 6A). In addition, the gut microbiota *Acinetobacter, Bacillus, Staphylococcus* were strongly positively correlated with water in stool, diarrhea score, proinflammatory factor; whereas they were negatively correlated with tight junctional proteins, mucus proteins, and suppressible inflammatory factor (Fig. 6B). On the contrary, the gut microbiota *Akkermansia, Oscillospira, Bacteroides,* and *Ruminococcus* were strongly negatively correlated with water in stool, diarrhea score,

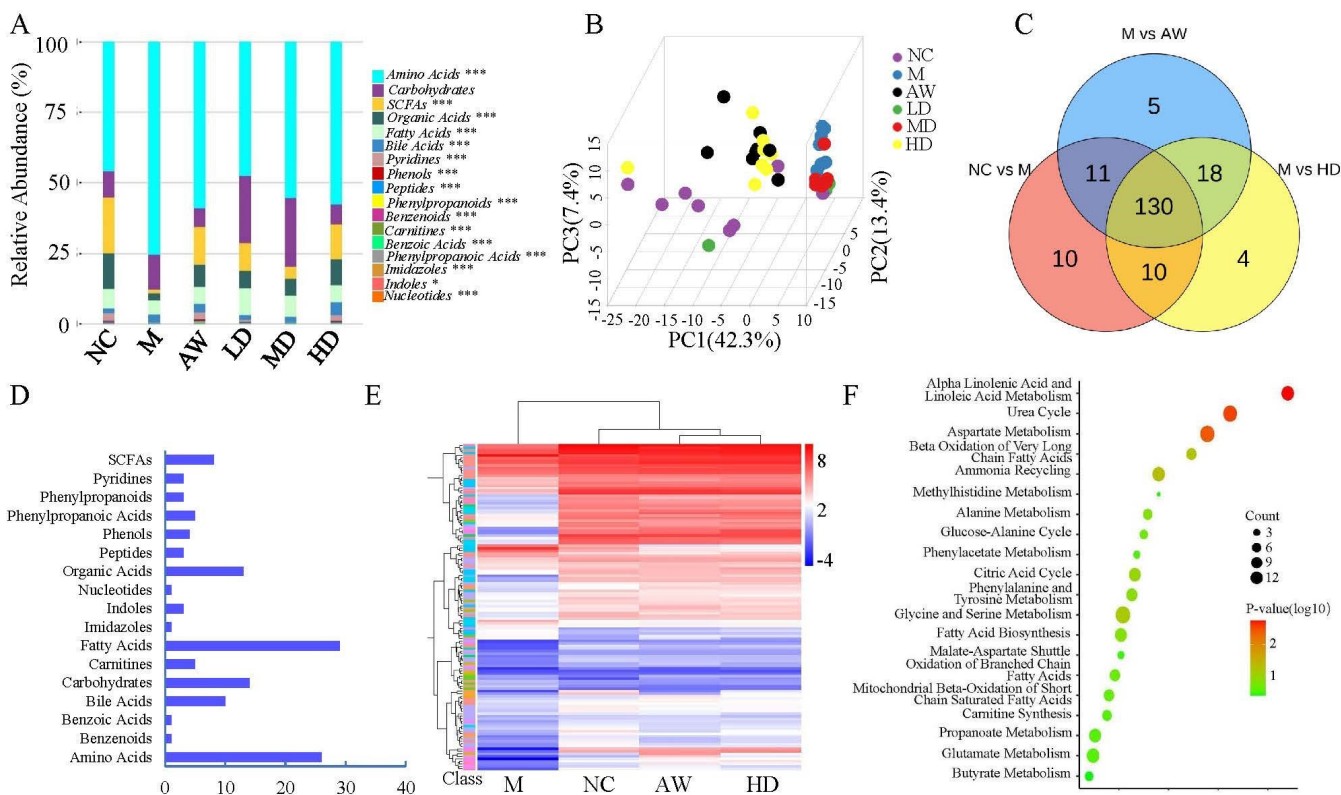

**FIG 5** *C. butyricum* RH2 partially restored the alteration of fecal metabolites in juvenile AAD mice. (A) The relative abundance of each metabolite class in different groups is shown in a stacked bar chart. The differential metabolites obtained using the Kruskal-Wallis test, ***$P < 0.001$; **$P < 0.01$; *$P < 0.05$. (B) PCA plots among the NC, M, AW, LD, MD, and HD groups. (C) Venn diagram between the three pairwise groups (NC vs M, M vs AW, and M vs HD) overlapped 130 significantly changed metabolites. (D) One hundred thirty metabolites were grouped into 17 classes; the X-axis represented the number of changed metabolites in the same class, and the Y-axis represented different classes. (E) The heatmap showed the cluster analysis of 130 overlapping metabolites. Kruskal-Wallis test, threshold value for differential metabolites selection is $P < 0.05$. (F) Pathway enrichment analysis bubble charts using pathway-associated metabolite sets (small molecule pathway database [SMPDB]).

proinflammatory factor, but were positively correlated with tight junctional proteins, mucus proteins, and suppressible inflammatory factor (Fig. 6B).

Interestingly, we also observed that most of the *C. butyricum*-restored gut microbiota were significantly correlated with *C. butyricum*-restored fecal metabolites (Fig. 7). The microbes and metabolites can be roughly divided into two correlation patterns. For example, *Acinetobacter, Bacillus,* and *Staphylococcus* were negatively correlated with some SCFAs, such as butyric acid, acetic acid, pyruvic acid, etc., and positively correlated with other fecal metabolites, such as lactulose, taurodeoxycholic acid (TDCA), tauroursodeoxycholic acid (TUDCA), etc. (Fig. 7). However, *Akkermansia, Oscillospira, Bacteroides,* and *Ruminococcus* showed opposite patterns to the aforementioned bacteria (Fig. 7).

In summary, the findings suggest that *C. butyricum* RH2 may ameliorate diarrhea symptoms associated with AAD through modulation of gut microbiota and related metabolites. Potential mechanisms include SCFA-mediated enhancement of intestinal barrier function, suppression of pathobionts, and regulation of bile acid metabolism pathways.

## DISCUSSION

In the present study, we established a juvenile mouse model of AAD using a combination of streptomycin, ampicillin, and clindamycin to simulate common broad-spectrum antibiotic exposure in pediatric clinics. The model successfully reproduced the main characteristics of human AAD, including diarrhea, microbiota diversity loss, and intestinal

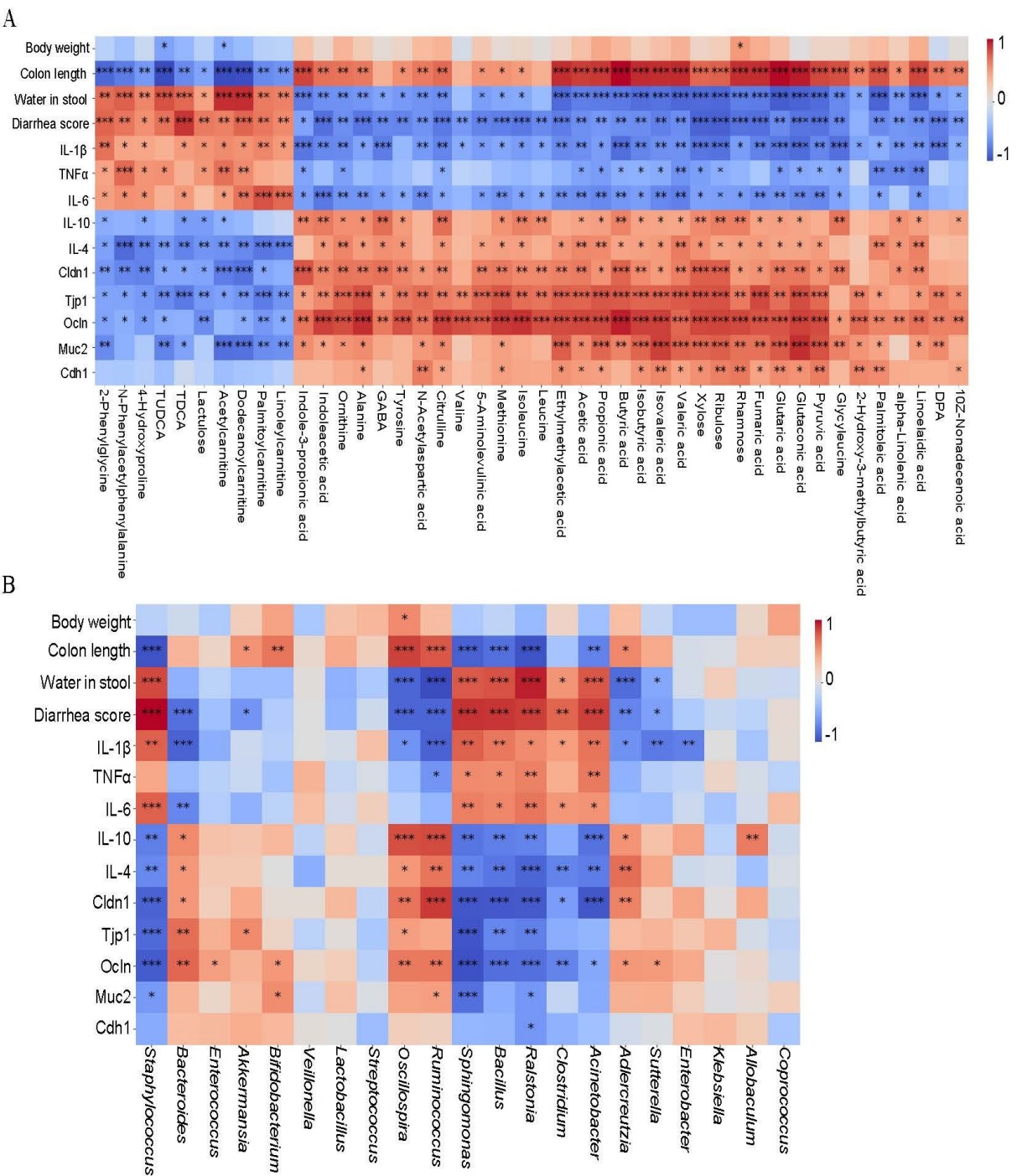

**FIG 6** Spearman's correlation analysis among *C. butyricum* RH2 reversed gut microbiota, fecal metabolites, and AAD indexes. (A) Correlation between *C. butyricum* reversed gut microbiota and AAD indexes, the X-axis represented gut microbiota, and the Y-axis represented AAD indexes. (B) Correlation between *C. butyricum* reversed fecal metabolites and AAD indexes, the X-axis represented metabolites, and the Y-axis represented AAD indexes. The red color showed the positive correlation, and the blue color showed the negative correlation, *P < 0.05, **P < 0.01, and ***P < 0.001.

barrier disruption. Notably, supplementation with *C. butyricum* RH2 alleviated antibiotic-induced diarrhea even during continued antibiotic administration, a clinically relevant scenario in pediatric care where antibiotic withdrawal is often not feasible. These findings collectively provide a practical foundation for evaluating *C. butyricum* RH2 as a potential adjunctive strategy in pediatric AAD management.

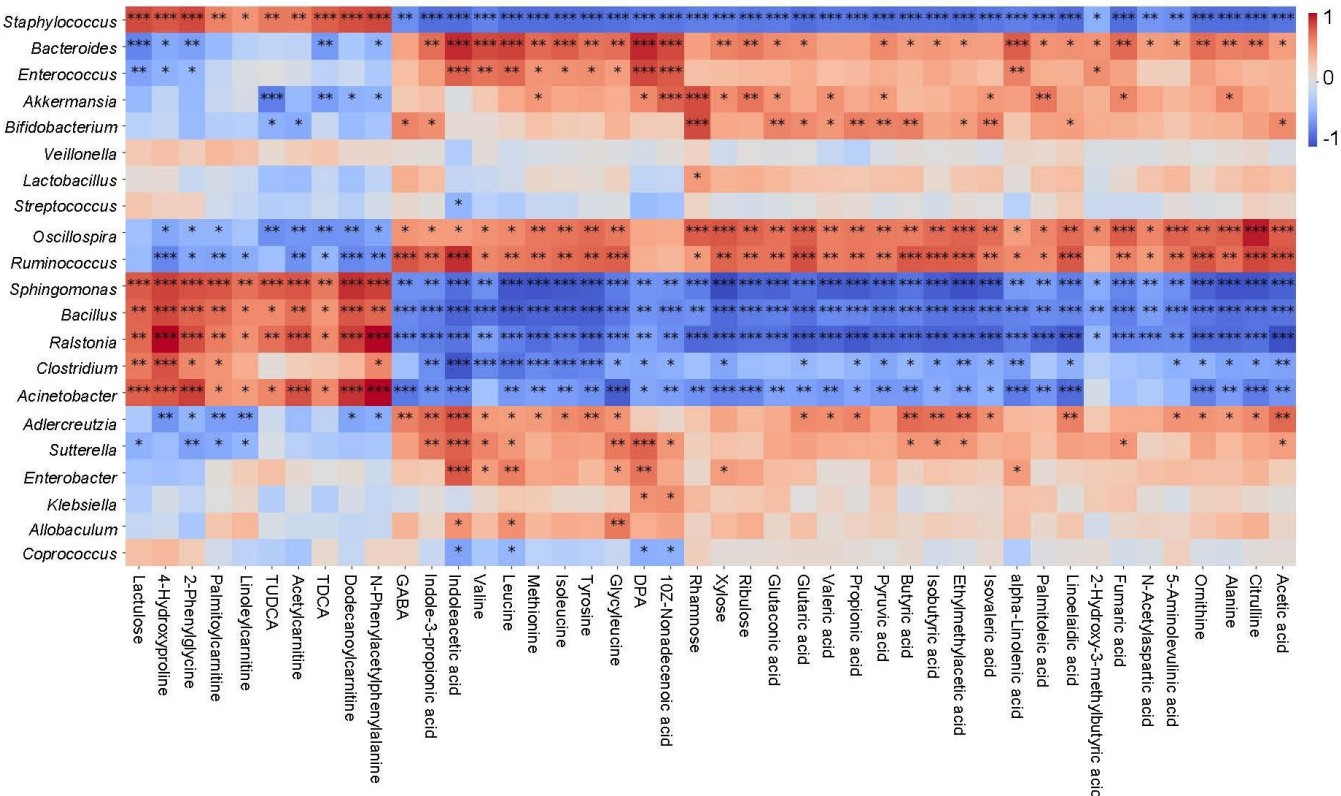

**FIG 7** Spearman's correlation analysis among gut microbiota and fecal metabolites. Correlation between *C. butyricum* reversed metabolites and gut microbiota, the X-axis represented metabolites, and the Y-axis represented gut microbiota. The red color showed the positive correlation, and the blue color showed the negative correlation, *$P < 0.05$, **$P < 0.01$, and ***$P < 0.001$.

Various bacteria in the gut microbiota constrain each other through occupying effects, nutritional competition, metabolic mutualism, production of organic acids and antibacterial substances, and immune synergy mechanisms, forming a balance between the microbiota in a certain composition and proportion (25). Antibiotics significantly disrupt both the composition and functional attributes of the gut microbiome, markedly reduces markedly reducing microbial diversity, promoting the overgrowth of opportunistic taxa, such as *Acinetobacter* and *Bacillus*, and diminishing beneficial commensals, including *Bacteroides*, *Oscillospira*, and *Akkermansia*. These alterations impair colonization resistance and epithelial stability and may persist well after antibiotic withdrawal (13). Similarly, the withdrawal of antibiotics for 14 days did not restore the fecal microbial diversity and richness of AAD juvenile mice in the present study (Fig. S2). RH2 supplementation partially reversed this dysbiosis, promoting the re-expansion of butyrate-producing genera (*Ruminococcus*, *Oscillospira*) and mucin-associated bacteria (*Akkermansia*). These shifts align with known microbial functions in SCFA production, mucus maintenance, and epithelial barrier support, suggesting that RH2 may facilitate recovery through microbiota-dependent pathways.

Changes in community structure alone rarely explain improvements in epithelial function. We next explored whether the recovered taxa participate in metabolic pathways connected to barrier repair. This transition is essential for understanding the biological meaning of the observed ecological changes. Beneficial genera, such as *Akkermansia* and *Oscillospira,* are well-known contributors to SCFA production and mucus maintenance, and in our study, they displayed positive correlations with barrier-related genes, including *Ocln*, *Cldn1*, *Cdh1*, and *Muc2*. Prior research has shown that giving butyrate and propionate is an established inducer of epithelial tight-junction proteins and mucin synthesis (26, 27). This evidence provides a mechanistic bridge

linking RH2-associated microbial restoration to the observed improvements in epithelial barrier function. Together, these patterns suggest a plausible microbiota–metabolite–epithelium axis. Nevertheless, these relationships remain associative, and a causal link cannot yet be confirmed.

Metabolomic profiling further supports these observations. RH2 treatment partially restored SCFA levels and altered several other metabolic pathways, including bile acid and fatty acid metabolism. These metabolic shifts occurred alongside reductions in inflammatory cytokines and improvements in epithelial morphology. The observed patterns are consistent with the well-established roles of SCFAs as epithelial energy sources and signaling molecules that activate G-protein-coupled receptors (GPR43, GPR109A) while inhibiting histone deacetylase activity, thereby enhancing tight-junction stability and anti-inflammatory responses (28–30). Antibiotics also increased conjugated bile acids, such as TDCA and TUDCA, which have been implicated in epithelial permeability and diarrheal fluid loss (31). Restoration of microbial bile acid transformation pathways may rebalance farnesoid X receptor and Takeda G-protein receptor 5 signaling, contributing to mucosal immune homeostasis (32, 33). Moreover, perturbations in linoleic and α-linolenic acid pathways observed in AAD are associated with impaired peroxisome proliferator-activated receptor activation and enhanced intestinal inflammation (34, 35). RH2 supplementation reversed these alterations, orchestratings a coordinated restoration of microbial and host metabolic networks, aligning with known protective pathways against antibiotic-associated epithelial injury (3). Integrating these metabolomic findings provides a more comprehensive mechanistic context, linking RH2-associated microbial restoration with host metabolic and barrier responses.

In summary, our findings suggest that *C. butyricum* RH2 represents a rational adjunct candidate for pediatric AAD management, coordinating the partial restoration of microbial composition, metabolite balance, and epithelial integrity under continuous antibiotic exposure. However, the mechanistic implications presented here remain exploratory. Because *C. butyricum* RH2 is a transient species and no metabolite supplementation or receptor-blocking assays were performed, direct causal pathways cannot yet be established. Future studies incorporating SCFA supplementation, SCFA receptor inhibition, time-series multi-omics, and post-treatment washout experiments will be essential to validate the mechanistic hypotheses implicated by our correlation analyses and to determine the durability of RH2-mediated benefits.

## ACKNOWLEDGMENTS

Y.Z., K.G., and Y.S. performed the experiments, analyzed data, prepared figures, and wrote the manuscript; F.G. carried out the 16S rRNA-sequencing and analysis; S.C. and Q.R. participated in experiments; Y.W. contributed to the study concept, design, and revised the manuscript.

## AUTHOR AFFILIATION

[1]Hangzhou Grand Biologic Pharmaceutical Inc., Hangzhou, China

## AUTHOR ORCIDs

Yufeng Zhao  http://orcid.org/0009-0003-3753-8381
Yu Wang  http://orcid.org/0009-0002-5568-5425

## AUTHOR CONTRIBUTIONS

Yufeng Zhao, Conceptualization, Data curation, Formal analysis, Methodology, Visualization, Writing – original draft, Writing – review and editing | Kan Gao, Conceptualization, Supervision, Writing – original draft, Writing – review and editing | Yunlong Shang, Investigation, Project administration, Supervision, Writing – review and editing | Shen Cheng, Data curation, Investigation, Methodology | Qianlei Ren, Data curation,

Investigation, Methodology, Visualization | Fengyu Guo, Formal analysis, Visualization | Yu Wang, Conceptualization, Project administration, Resources, Supervision

## DATA AVAILABILITY

Sequencing data sets have been deposited to the NCBI Sequence Read Archive under BioProject accession number PRJNA1157009.

## ETHICS APPROVAL

All experimental protocols complied with the National Institutes of Health Guide for the Care and Use of Laboratory Animals and were approved by the Institutional Animal Care and Use Committee of Hangzhou Qingda Kerui Biotechnology Co., Ltd. (approval no. QDAE20230625001).

## ADDITIONAL FILES

The following material is available online.

### Supplemental Material

**Supplemental material (Spectrum01976-25-s0001.docx).** Table S1; Fig. S1 to S3.

### Open Peer Review

**PEER REVIEW HISTORY (review-history.pdf).** An accounting of the reviewer comments and feedback.

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
