## [Reviewer comments · Microbiology Spectrum]

Microbiology Spectrum

***Clostridium butyricum* RH2 ameliorates Diarrhea in juvenile mice under continuous antibiotics exposure by modulating gut microbiota and metabolome**

Yufeng Zhao, Kan Gao, Yunlong Shang, Shen Cheng, Qianlei Ren, Fengyu Guo, and Yu Wang

Corresponding Author(s): Yu Wang, Hangzhou Grand Biologic Pharmaceutical Inc.

Review Timeline:

Submission Date:	July 2, 2025
Editorial Decision:	September 10, 2025
Revision Received:	September 25, 2025
Editorial Decision:	November 11, 2025
Revision Received:	November 18, 2025
Accepted:	December 6, 2025

Editor: Yuan Pin Hung

Reviewer(s): Disclosure of reviewer identity is with reference to reviewer comments included in decision letter(s). The following individuals involved in review of your submission have agreed to reveal their identity: Sahar M.Jawad Abduladheem (Reviewer #1)

Transaction Report:

DOI: <https://doi.org/10.1128/spectrum.01976-25>

Re: Spectrum01976-25 (*Clostridium butyricum* RH2 ameliorates Diarrhea in juvenile mice under continuous antibiotics exposure by modulating gut microbiota and metabolome)

Dear Ms. Yufeng Zhao:

Thank you for the privilege of reviewing your work. Below you will find my comments, instructions from the Spectrum editorial office, and the reviewer comments.

Revision Guidelines

Sincerely,
Yuan Pin Hung
Editor
Microbiology Spectrum

Reviewer #1 (Comments for the Author):

From my perspective of view, in this manuscript:

Line: 36 Repetition: the phrases "young mice" and "juvenile mice" appeared twice; the term can be unified.

Line 45: The flow between the sentence describing the results and the concluding sentence can be improved (the last sentence appears more appended than integrated).

Line 151 : The reference (housekeeping) gene used for normalizing gene expression (such as GAPDH or β -actin) was not specified.

Line 153: The reaction volume, RNA concentration, and thermal cycling conditions for qPCR were not specified, and these are often required.

Line 221: For better flow: the transition from describing the experiment to presenting the results (weight loss, lethargy) was abrupt; it would be preferable to add a linking sentence.

Line 369: Given that SCFAs and bile acids play central roles in gut homeostasis, how can we determine whether the observed therapeutic effect of *C. butyricum* RH2 is primarily driven by SCFA production, bile acid modulation, or synergistic interactions between both pathways?

Line 434: Given that *C. butyricum* is a transient rather than colonizing species, to what extent can its long-term ecological and therapeutic effects be sustained after discontinuation of supplementation, and how might this impact its clinical applicability in AAD treatment?

Reviewer #2 (Comments for the Author):

- This study investigates the therapeutic potential of the probiotic strain *Clostridium butyricum* RH2 in alleviating antibiotic-associated diarrhea (AAD) in juvenile mice. The authors developed a murine model using a triple-antibiotic cocktail and evaluated the effects of *C. butyricum* RH2 on gut microbiota composition, intestinal barrier integrity, inflammation, and fecal metabolomics. The study includes multiple treatment groups (low, medium, high doses of RH2, antibiotic withdrawal, and controls).
- It uses well-established methods for assessing diarrhea severity, histology, gene expression, microbiota profiling, and metabolomics. Here are some suggestions to improve the article;

1. Typographical Errors:

- "therap eutic" → should be "therapeutic"
- "calcu lated" → should be "calculated"
- "significan ce" → should be "significance"
- "observ ed" → should be "observed"
- "chan ged" → should be "changed"

2. Inconsistent Spacing and Hyphenation:

- "anti - inflammatory" → should be "anti-inflammatory"
- "co - administration" → should be "co-administration"
- "microbiota - derived" → should be "microbiota-derived"

3. Redundant or Awkward Phrasing:

- "Given the heightened clinical vulnerability of pediatric populations to AAD complications..." → could be simplified for clarity.
- "This persistent deficit highlights the long - term detrimental effects..." → "persistent deficit" is vague; consider specifying the deficit.

4. Subject-Verb Agreement:

- "The microbes and metabolites could be roughly divided..." → "can be" is more appropriate in scientific writing unless referring to past analysis.

5. Improper Use of Articles:

- "a Gram - positive, obligate anaerobic bacterium" → correct, but sometimes "a" is missing before similar phrases.

Technical Mistakes and Clarity Issues

1. Ambiguous Dose Descriptions:

- "low (LD: 1.05×10^5 CFU daily)" → consider clarifying whether this is per mouse or per group.

2. Inconsistent Terminology:

- "fecal water content" and "water in stool" are used interchangeably; standardize terminology.

3. Missing Units or Definitions:

- "Figure 1B" mentions body weight but doesn't specify units (grams?).

4. Overuse of Passive Voice:

- "Samples were immediately weighed..." → consider active voice for clarity: "We immediately weighed the samples..."

5. Scientific Jargon Without Explanation:

- Terms like "LEfSe analysis," "Faith's PD," and "SMPDB" are used without brief definitions, which may confuse non-specialist readers.

Reviewer #3 (Public repository details (Required)):

16S rRNA gene sequence.

Reviewer #3 (Comments for the Author):

Dear Authors,

Thanks for your work.

Line 75: It is important to highlight *Clostridium* spp is not yet considered GRAS for probiotic applications unlike *Bifidobacterium* and *Lactobacillus* that have been widely studied and accepted as GRAS.

At the end of the discussion section, please include a paragraph describing the limitations of this study and possible strategies to address them in future studies.

From my perspective of view, in this manuscript:

Line 36 : Repetition: the phrases "young mice" and "juvenile mice" appeared twice; the term can be unified.

Line 45: The flow between the sentence describing the results and the concluding sentence can be improved (the last sentence appears more appended than integrated).

Line 151 : The reference (housekeeping) gene used for normalizing gene expression (such as GAPDH or β -actin) was not specified.

Line 153: The reaction volume, RNA concentration, and thermal cycling conditions for qPCR were not specified, and these are often required.

Line 221: For better flow: the transition from describing the experiment to presenting the results (weight loss, lethargy) was abrupt; it would be preferable to add a linking sentence.

Line 369: Given that SCFAs and bile acids play central roles in gut homeostasis, how can we determine whether the observed therapeutic effect of *C. butyricum* RH2 is primarily driven by SCFA production, bile acid modulation, or synergistic interactions between both pathways?

Line 434: Given that *C. butyricum* is a transient rather than colonizing species, to what extent can its long-term ecological and therapeutic effects be sustained after discontinuation of supplementation, and how might this impact its clinical applicability in AAD treatment?"

***Clostridium butyricum* RH2 ameliorates Diarrhea in juvenile mice under**
**continuous antibiotics exposure by modulating gut microbiota and metabolome**

Yufeng Zhao¹, Yunlong Shang¹, Shen Cheng¹, Tianyue Guan¹, Fengyu Guo¹, Yu
Wang^{1,*}

¹Hangzhou Grand Biologic Pharmaceutical Inc., Hangzhou, 310000, China

*Corresponding author.

Yu Wang, PhD

Hangzhou Grand Biologic Pharmaceutical Inc., Hangzhou, 310000, China

Tel: 86-571-2802-0125

Fax: 86-571-2802-0125

E-mail: wangyu@hzydsw.cn

Word count: 5247

**Abbreviations**

Antibiotic-associated diarrhea: AAD; *Clostridium butyricum*: *C. butyricum*; Claudin-1:
Cldn1; Claudin-5: Cldn5; Occludin: Ocln; Tight junction protein 1: Tjp1; Cadherin 1:
Cdh1; Interleukin 6: Il6; Tumor necrosis factor α : Tnf α ; Interleukin 10: Il10;
Interleukin 1 beta: Il1 β ; Interleukin 4: Il4; Short-chain fatty acids: SCFAs

**Abstract:**

Antibiotic-associated diarrhea (AAD) is a self-limiting disorder triggered by
antibiotic therapy in pediatric populations. Although multiple probiotics are clinically
employed for AAD management, the therapeutic efficacy of *Clostridium butyricum*
(*C. butyricum*) in pediatric AAD and its underlying mechanisms remain poorly
characterized. This study aimed to establish a juvenile mice model of AAD and
investigate the therapeutic potential of oral *C. butyricum* administration in young mice
subjected to continuous antibiotics exposure. We systematically assessed pathological
changes in colonic tissue, colitis severity, intestinal epithelial barrier integrity, fecal
metabolomic profiles, and gut microbiota diversity. Our analysis demonstrates that *C.*
*butyricum* ameliorates intestinal inflammation, enhances barrier function by
modulating the gut microbiota and its metabolites, and significantly alleviates
diarrhea symptoms in juvenile AAD mice. Furthermore, this study reveals that *C.*
*butyricum* can tolerate continuous antibiotics exposure within the gastrointestinal tract
of young mice, providing a scientific rationale for its co-administration with
antibiotics.

**Keywords:** antibiotic-associated diarrhea; gut barrier function; *Clostridium butyricum*
RH2; gut microbiota; metabolomics.

**Introduction**

As the most frequent complication of antimicrobial treatment,
antibiotic-associated diarrhea (AAD) is defined as unexplained diarrheal episodes
temporally linked to antibiotic use. All antibiotic classes demonstrate diarrheagenic
potential, particularly broad-spectrum agents including penicillins, cephalosporins,
and clindamycin[1]. The clinical manifestations depend on encompassing antibiotic
class, dosage regimen, treatment duration, administration route, and combination
therapies[2]. While AAD affects all age demographics, pediatric populations exhibit
heightened susceptibility, with reported prevalence rates of 20-35% in children[3].
Epidemiological surveillance data indicate substantial variability across care settings:
US pediatric cohorts demonstrate 6% incidence in ambulatory care versus 80% in
hospitalized patients[3]. Chinese clinical studies, currently limited to inpatient settings,
report incidence rates ranging from 16.8% to 70.6%[4]. Current management remains
restricted to antibiotic discontinuation or substitution, with no targeted therapies
available. Pathophysiological mechanisms involve antibiotic-induced dysbiosis
characterized by gastrointestinal microbiota disruption, opportunistic pathogen
proliferation, and metabolic dysfunction.

Probiotics are defined as live microorganisms that confer health benefits to the
host when administered in adequate doses[5]. Substantial evidence from preclinical
and clinical studies has established that antibiotic-induced gut microbiota dysbiosis
plays a fundamental role in AAD pathogenesis. This mechanistic understanding
supports probiotics as viable therapeutic agents for microbiota restoration. Multiple
randomized controlled trials have demonstrated the efficacy of specific microbial
strains in AAD prevention, including
*Bifidobacterium* spp., *Clostridium* spp., *Lactobacillus* spp., and the fungal
species *Saccharomyces boulardii* [6-13].

*Clostridium butyricum* (*C. butyricum*), a strictly anaerobic spore-forming
bacillus, has been extensively utilized as a probiotic in Asian countries for decades.
This microorganism demonstrates ubiquitous distribution, inhabiting soil, dairy
products, vegetable matter, and the human gastrointestinal tract. As an early gut

colonizer, it is detected in 10-20% of adult populations[14]. Notably, *C. butyricum*
encompasses multiple strains exhibiting diverse health-promoting mechanisms
through host-microbe interactions[15]. Emerging evidence positions this species as a
multifunctional therapeutic agent, demonstrating efficacy in gastrointestinal,
neurological, and metabolic disorders, alongside potential anticancer properties[16].
However, the potential mechanism by which *C. butyricum* RH2 alleviates diarrhea
symptoms induced by multiple antibiotics in juvenile mice has not been explored.

In this study, we employed *C. butyricum* RH2 strain to investigate its therapeutic
efficacy and mechanistic underpinnings in a juvenile murine model of
antibiotic-associated diarrhea (AAD). Given the heightened clinical vulnerability of
pediatric populations to AAD complications, we established a developmentally
relevant mouse model using 4-week-old C57BL/6J mice. The model was induced
through sequential oral administration of an antibiotic cocktail (ampicillin,
streptomycin, and clindamycin), simulating pediatric clinical conditions. Our findings
demonstrate that *C. butyricum* RH2 exhibits significant potential as a probiotic
adjuvant for maintaining intestinal homeostasis in antibiotic-dependent pediatric AAD
management.

**Materials and Methods**

**Reagents and Probiotics Strain**

The juvenile AAD murine model was established through daily gavage
administration of an antibiotic cocktail containing clindamycin (Jiudian, Hunan,
China), ampicillin (United Laboratories, Hong Kong, China), and streptomycin
(Lukang, Shandong, China) dissolved in normal saline. Antibiotic cocktail
administration was performed daily at 10:00.

*C. butyricum* RH2 lyophilized powder (Grand Biologic Pharmaceutical; Batch
J202304072) was reconstituted in sterile 0.9% NaCl solution to achieve target
concentrations. A minimum 4-hour interval was maintained between probiotic and
antibiotic administrations to prevent pharmacological interference.

**Animal Experimental Design**

Sixty specific pathogen-free (SPF) male C57BL/6J mice (3-week-old) were
obtained from Shanghai SLAC Laboratory Animal Co., Ltd. (Shanghai, China). The
animals were maintained under controlled environmental conditions (temperature: 22
\pm 2°C; humidity: 40 \pm 5%) with a 12-hour light/dark cycle, and provided ad libitum
access to food and water. All experimental protocols complied with the National
Institutes of Health Guide for the Care and Use of Laboratory Animals and were
approved by the Institutional Animal Care and Use Committee of Hangzhou Qingda
Kerui Biotechnology Co., Ltd. (Approval No.: QDAE20230625001).

As illustrated in Figure 1A, following a 7-day acclimatization period, mice were
randomly allocated into six experimental groups: normal control (NC),
antibiotic-associated diarrhea model (M), antibiotic withdrawal control (AW), and
three *C. butyricum* RH2 treatment groups receiving low (LD: 1.05×10^5 CFU daily),
medium (MD: 1.05×10^7 CFU daily), or high doses (HD: 1.05×10^9 CFU daily).
Based on established protocols demonstrating successful AAD model induction
through 3-day antibiotic administration [17], therapeutic interventions commenced
immediately post-antibiotic treatment and continued for 14 days. To prevent
spontaneous recovery reported in untreated AAD models[17], antibiotic
administration was maintained throughout the treatment phase. The AW group served
as a self-recovery control, receiving antibiotics for 3 days followed by 14 days of
saline treatment. Fecal samples were collected from all groups for subsequent
analyses.

**Diarrhea Assessment**

Diarrhea severity was quantitatively assessed through fecal consistency scoring
and fecal moisture content analysis following established methodology[18].
Antibiotic-associated diarrhea status was classified based on the following visual
grading scale: (1) Alert behavioral status with formed, the feces are elliptical in shape,
with a hard texture and a brownish color, score = 1; (2) General mental state, the feces
are sausage shaped, with a smooth surface and a yellowish color, score = 2; and (3)
Bad mental state, the feces are in the shape of sausages or paste, with a wet and soft
texture, and a yellowish color, score =3. Fresh fecal samples were collected over a

standardized 1-hour period using metabolic cages. Samples were immediately
weighed (wet weight), desiccated at 60°C to constant mass, and reweighed (dry
weight). Fecal moisture content was calculated using:

$$144 \quad \text{Fecal Water Content (\%)} = [(\text{Wet Weight} - \text{Dry Weight})/\text{Wet Weight}] \times 100$$

**RNA extraction and RT-PCR**

Total RNA was isolated from tissue specimens using TRIzol™ Reagent (Thermo
Fisher Scientific, Waltham, MA, USA) following the manufacturer's protocol.
First-strand cDNA synthesis was performed with the HiScript III Reverse
Transcriptase Kit (Vazyme Biotech Co., Ltd., Nanjing, China). Quantitative real-time
polymerase chain reaction (qPCR) analyses were conducted on a QuantStudio™ 7500
Fast Real-Time PCR System (Applied Biosystems, Foster City, CA, USA) using
ChamQ Universal SYBR qPCR Master Mix (Vazyme Biotech Co., Ltd.), as
previously described[19]. The primer sequences are shown in Supplementary Table 1.

**Histological examination**

For histological analysis, colon tissue samples were fixed in 4%
paraformaldehyde and embedded in paraffin. Hematoxylin and eosin (H&E) staining
was performed according to established protocols[19].

**16S rRNA gene amplicon Sequencing**

Extract bacterial DNA from mice feces were obtained using OMEGA Soil DNA
Kit (M5635-02; OMEGA Bio-Tek, Norcross, GA, USA). NanoDrop NC2000
spectrophotometer (Thermo Fisher Scientific, Waltham, MA, USA) followed by
agarose gel electrophoresis were used to measure the quality of extracted DNA. PCR
amplification was performed on the V3-V4 region of the bacterial 16S rRNA gene
using forward primer 338F (5'-ACTCCTACGGGAGGCAGCA-3') and reverse
primer 806R (5' -GGACTACHVGGGTWTCTAAT-3'). Vazyme VAHTSTM DNA
Clean Beads (Vazyme, Nanjing, China) was used to purify PCR amplicons, which
quantifying with QuantiT PicoGreen dsDNA assay kit (Invitrogen, Carlsbad, CA,
USA). Purified amplicons were merged in equal quantities and paired for sequencing
using Illumina NovaSeq platform and NovaSeq6000 SP kit (500 cycles). The

sequence data were processed using QIIME2 according to previously described
methods[20].

**Metabolomics analysis**

Metabolites in fecal samples were quantified using an ultra-performance liquid
chromatography-tandem mass spectrometry (UPLC-MS/MS) system (ACQUITY
UPLC-Xevo TQ-S; Waters Corporation, Milford, MA, USA) with the Q300
Metabolite Array Kit (Metabo-Profile Biotechnology, Shanghai, China). Raw data
files generated by UPLC -MS/MS were processed using iMAP platform (v1.0;
Metabo-Profile, China). Multivariate statistical analysis (ANOVA or Kruskal-Wallis
test, selected based on data normality and variance homogeneity) was applied to
perform principal component analysis (PCA) and identify differential metabolites
between groups. Differential metabolites were defined by a significance threshold of
$P < 0.05$. Shared metabolites among pairwise comparisons (NC vs. M, M vs. AW, and
M vs. HD) were visualized using Venn diagrams, while hierarchical clustering
analysis illustrated metabolite expression patterns across groups. Metabolite
enrichment analysis for *C. butyricum* RH2-reversed metabolic alterations was
conducted through the MetaboAnalyst platform.

**Statistics**

All data were expressed as the mean \pm standard error of the mean (SEM).
One-way ANOVA among groups followed by Student's t-test between groups were
performed to analyze the differences using the Prism 9.0 program (GraphPad
Software, San Diego, Canada). Spearman correlation analyses between gut microbiota
composition, fecal metabolites, and antibiotic-associated diarrhea (AAD) indices were
performed using SPSS version 24 (IBM Corp., Armonk, NY, USA). Statistical
significance was defined as $p < 0.05$.

**Data Availability Statement**

Sequencing datasets have been deposited to the NCBI Sequence Read Archive
under BioProject accession numbers PRJNA1157009.

**Disclosure of potential conflicts of interest**

The authors disclose the following potential conflicts of interest: Y.Z., Y.S., S.C.,

[revised manuscript text omitted]

Antibiotic-associated diarrhea (AAD) is widely recognized as a common adverse
reaction to antibiotic therapy, particularly in pediatric populations. Current pediatric
AAD management guidelines recommend antibiotic withdrawal as a primary
therapeutic strategy, provided that discontinuation does not compromise treatment of

the underlying infection. To investigate the pathophysiology of pediatric AAD, we
developed a juvenile murine model using C57BL/6J mice administered with
broad-spectrum antibiotics, which reliably induced diarrheal symptoms characteristic
of clinical AAD. *C. butyricum* RH2, a Gram-positive, obligate anaerobic bacterium
within Clostridium cluster I, demonstrates significant butyrogenic capacity[16, 21].
Butyrate-producing microbiota are increasingly recognized for their
anti-inflammatory properties through epithelial barrier reinforcement and immune
modulation[22]. Our investigation revealed that supplementing *C. butyricum* could
effectively alleviate AAD-related symptoms, as evidenced by water in stool and
diarrhea score diminished (Figure 1), colon inflammation ameliorated (Figure 2).
Furthermore, we confirmed that *C. butyricum* RH2 can modulate the gut microbiota
and metabolites, protected gut barrier function and improved gut microbiota
homeostasis, which also helps to improve the symptoms of pathological diarrhea.
Notably, therapeutic outcomes including diarrhea severity (Figure 1C), colonic
shortening (Figure 2B), and intestinal barrier parameters (Figure 3) demonstrated
comparable efficacy between continuous antibiotic therapy with *C. butyricum* RH2
supplementation and complete antibiotic withdrawal. This finding suggests that
concurrent administration of *C. butyricum* RH2 with essential antibiotic therapy may
prevent AAD development in pediatric patients requiring prolonged antimicrobial
treatment.

The gastrointestinal tract serves as the principal colonization site for human
microbiota, harboring 1000 types of approximately 10^{13} - 10^{14} bacteria residing in the
normal human gut[23]. Various bacteria in the gut microbiota constrain each other
through occupying effects, nutritional competition, metabolic mutualism, production
of organic acids and antibacterial substances, and immune synergy mechanisms,
forming a balance between the microbiota in a certain composition and proportion[24].
Antibiotics cause significant disruptions to the normal composition and functional
attributes of the gut microbiome, such alterations can persist well after the cessation
of antibiotic administration[25]. Similarly, the withdraw of antibiotics for 14 days did

not restore the fecal microbial diversity and richness of AAD juvenile mice in the
present study (Supplementary Figure 2). Dysbiosis of the microbiota induced by
antibiotics represent that inhibition of sensitive microbiota in the intestine, leading to
excessive proliferation of non-sensitive microbiota, decreased diversity of gut
microbiota, and profound changes in the composition and proportion of
microbiota[25]. Thus, the gut microbiota might play a critical role in the pathogenesis
and progression of AAD.

The 16S rRNA gene sequencing approach, involving PCR amplification of
hypervariable regions (V3-V5) from fecal samples followed by Illumina deep
sequencing, represents the predominant methodology for investigating *C. butyricum*'s
impact on gut microbiota composition[26]. Current research presents conflicting
findings regarding microbial community responses to *C. butyricum* supplementation.
While multiple studies report significant alterations in α -diversity indices and
β -diversity patterns[27-32], other investigations, including our current work
(Supplemental Figure 2), demonstrate no statistically significant changes in these
microbial community metrics [33-35]. These discrepancies may stem from
methodological variations in 16S rRNA sequencing protocols and multiple
confounding factors affecting gut microbiota profiles, including host genotype,
environmental conditions (housing/litter), dietary composition, and vendor-specific
microbiome signatures [36-38].

As a transient commensal bacterium, *C. butyricum* exhibits unique ecological
characteristics in the gastrointestinal tract. This non-colonizing species demonstrates
synergistic relationships with beneficial gut microbiota through trophic interactions
while simultaneously suppressing pathogenic and putrefactive bacterial populations
via competitive exclusion and metabolic inhibition. These ecological patterns are
corroborated by our 16S rRNA sequencing analyses, revealing that high-dose *C.*
*butyricum* RH2 administration significantly elevated the relative abundance of
beneficial genera (*Ruminococcus*, *Oscillospira*, *Bacteroides*, and *Akkermansia*) while
suppressing pathogenic taxa (*Acinetobacter*, *Staphylococcus*, and *Pseudomonas*). This
microbial modulation aligns with established clinical correlations - a European

population study demonstrated that *Oscillospira* abundance positively associates with
stool firmness and negatively correlates with diarrheal status[39]. Our findings
substantiate this relationship through a significant inverse correlation between fecal
water content and *Oscillospira* abundance (Figure 6B).

Accumulating evidence underscores the critical role of intestinal epithelial
barrier integrity in the pathogenesis of various intestinal and systemic disorders[40].
Studies have shown that the use of antibiotics can lead to barrier dysfunction or
“intestinal leakage”[41]. A key characteristic of *A. muciniphila* lies in its capacity to
degrade intestinal mucin glycoproteins through fermentation processes, generating
sulfate metabolites along with short-chain fatty acids including acetate and
propionate. This mucolytic activity facilitates mucosal layer renewal and structural
reinforcement, thereby improving barrier function and limiting microbial
translocation[42]. In this study, we found a thinner, penetrable mucus layer and a
rapid reduction in the expression of *Muc2*, *Cdh1*, *Ocln*, *Cldn1*, *Cldn5* and *Tjp1* in the
AAD mice, while *C. butyricum* RH2 treatment alleviated antibiotic-induced barrier
dysfunction (Figure 3). These results suggested that the reduced severity of diarrhea
in the supplementing the *C. butyricum* RH2 group may be attributed to the protection
of gut barrier integrity (Figure 3). Consistently, the feces of AAD mice supplemented
with *C. butyricum* RH2 showed a significant increase in the genus *Akkermansia* in our
study (Figure 4F), and the correlation analysis also suggested that *Akkermansia* was
positively linked to tight junction protein gene expression but negatively associated
with diarrhea-related index and proinflammatory gene expression(Figure 6A),
indicating that *C. butyricum* RH2 improved the intestinal barrier function and
alleviated diarrhea symptoms in AAD mice by upregulating the genus *Akkermansia*.
Taken together, our study confirmed that probiotic *C. butyricum* RH2 was able to
improve the gastrointestinal symptoms of AAD and induce *Oscillospira* and
*Akkermansia* to gradually become the dominant species in the gut.

Mounting evidence indicates that gut microbiota-derived metabolites play a
pivotal role in modulating the pathogenesis of metabolic disorders[43-45].
Antibiotic-induced gut microbiota remodeling has been shown to significantly alter

intestinal metabolomic profiles[46]. AAD is typically characterized by decreased
intestinal short chain fatty acids (SCFAs) concentrations, accumulation of luminal
carbohydrates and colonic bile acids, altered water absorption, and ultimately
diarrhea[47]. Our metabolomic analysis identified several fecal metabolites including
short-chain fatty acids (SCFAs), bile acids (BAs), amino acids (AAs), etc. Short chain
fatty acids(SCFAs) are rapidly absorbed by the colon and stimulate Na-dependent
fluid absorption via a cyclic AMP-independent process with Na-H, SCFA-HCO₃, and
Cl-SCFAs exchanges, which means the reduction levels of SCFAs might lead to AAD
because of the NaCl and water absorption was damaged [48, 49]. More and more
probiotics are reported that can directly promote intestinal SCFAs by producing
organic acids, lactic acid, and acetic acid, or providing a more suitable environment
for SCFAs producing bacteria[50, 51]. Our research also confirms that supplementing
*C. butyricum* RH2 can effectively elevates fecal SCFA concentrations (Figure 5).

Bile acids (BAs) are the metabolic products of dietary substrates and play a vital
role in metabolism and immunological regulation[52, 53]. Approximately 95% of
luminal bile acids are reabsorbed in the distal ileum in healthy individuals, the
remaining amounts are modified by intestinal bacteria and then are either excreted or
passively absorbed[54]. Excessive secretion of bile acids or insufficient absorption of
bile acids lead to surplus bile acids present in the intestine, which increasing the
permeability of the intestinal mucosa and affecting the absorption of water and
sodium in the intestine, ultimately resulting in a large amount of intestinal fluid and
causing diarrhea[52, 53]. Our findings corroborate this mechanism, demonstrating
significant positive correlations between two specific bile acids—taurodeoxycholic
acid (TDCA) and tauroursodeoxycholic acid (TUDCA)—and key AAD clinical
markers, including fecal water content and diarrhea severity scores (Figure 6A).

Overall, we successfully established a novel antibiotic-associated diarrhea (AAD)
model in 4-week-old juvenile mice using a triple-antibiotic combination
(streptomycin, ampicillin, and clindamycin). Under continuous antibiotic exposure, *C.*
*butyricum* RH2 conferred intestinal protection by maintaining gut homeostasis
through modulation of microbiota composition and metabolites, enhancement of

intestinal epithelial barrier integrity, and suppression of colitis. Furthermore, this
study provides nonclinical evidence supporting the potential application of probiotic
*C. butyricum* to alleviate diarrhea in pediatric patients requiring continuous antibiotic
therapy.

**References**

- [1] L.V. McFarland, Antibiotic-associated diarrhea: epidemiology, trends and
treatment, *Future Microbiol* 3(5) (2008) 563-78.
- [2] C.L.R. Abad, N. Safdar, A Review of *Clostridioides difficile* Infection and
Antibiotic-Associated Diarrhea, *Gastroenterol Clin North Am* 50(2) (2021) 323-340.
- [3] L.V. McFarland, M. Ozen, E.C. Dinleyici, S. Goh, Comparison of pediatric and
adult antibiotic-associated diarrhea and *Clostridium difficile* infections, *World J*
*Gastroenterol* 22(11) (2016) 3078-104.
- [4] Q. Yang, Z. Hu, Y. Lei, X. Li, C. Xu, J. Zhang, H. Liu, X. Du, Overview of
systematic reviews of probiotics in the prevention and treatment of
antibiotic-associated diarrhea in children, *Front Pharmacol* 14 (2023) 1153070.
- [5] C. Hill, F. Guarner, G. Reid, G.R. Gibson, D.J. Merenstein, B. Pot, L. Morelli, R.B.
Canani, H.J. Flint, S. Salminen, P.C. Calder, M.E. Sanders, Expert consensus
document. The International Scientific Association for Probiotics and Prebiotics
consensus statement on the scope and appropriate use of the term probiotic, *Nat Rev*
*Gastroenterol Hepatol* 11(8) (2014) 506-14.
- [6] M. Storr, A. Stengel, Systematic review: clinical evidence of probiotics in the
prevention of antibiotic-associated diarrhoea, *MMW Fortschr Med* 163(Suppl 4)
(2021) 19-26.
- [7] Q. Guo, J.Z. Goldenberg, C. Humphrey, R. El Dib, B.C. Johnston, Probiotics for
the prevention of pediatric antibiotic-associated diarrhea, *Cochrane Database Syst*
*Rev* 4(4) (2019) Cd004827.
- [8] H.B. Xu, R.H. Jiang, H.B. Sheng, Meta-analysis of the effects of *Bifidobacterium*
preparations for the prevention and treatment of pediatric antibiotic-associated
diarrhea in China, *Complement Ther Med* 33 (2017) 105-113.
- [9] H. Szajewska, M. Kołodziej, Systematic review with meta-analysis:
*Saccharomyces boulardii* in the prevention of antibiotic-associated diarrhoea, *Aliment*
*Pharmacol Ther* 42(7) (2015) 793-801.
- [10] H. Szajewska, M. Kołodziej, Systematic review with meta-analysis:
*Lactobacillus rhamnosus* GG in the prevention of antibiotic-associated diarrhoea in

children and adults, *Aliment Pharmacol Ther* 42(10) (2015) 1149-57.

[11] H. Szajewska, M. Ruszczyński, A. Radzikowski, Probiotics in the prevention of
antibiotic-associated diarrhea in children: a meta-analysis of randomized controlled
trials, *J Pediatr* 149(3) (2006) 367-372.

[12] H. Szajewska, R.B. Canani, A. Guarino, I. Hojsak, F. Indrio, S. Kolacek, R. Orel,
R. Shamir, Y. Vandenplas, J.B. van Goudoever, Z. Weizman, Probiotics for the
Prevention of Antibiotic-Associated Diarrhea in Children, *J Pediatr Gastroenterol*
*Nutr* 62(3) (2016) 495-506.

[13] B.C. Johnston, J.Z. Goldenberg, P.O. Vandvik, X. Sun, G.H. Guyatt, Probiotics
for the prevention of pediatric antibiotic-associated diarrhea, *Cochrane Database Syst*
*Rev* (11) (2011) Cd004827.

[14] K.C. Mountzouris, A.L. McCartney, G.R. Gibson, Intestinal microflora of human
infants and current trends for its nutritional modulation, *Br J Nutr* 87(5) (2002)
405-20.

[15] N. Cassir, S. Benamar, B. La Scola, *Clostridium butyricum*: from beneficial to a
new emerging pathogen, *Clin Microbiol Infect* 22(1) (2016) 37-45.

[16] H. Seki, M. Shiohara, T. Matsumura, N. Miyagawa, M. Tanaka, A. Komiyama, S.
Kurata, Prevention of antibiotic-associated diarrhea in children by *Clostridium*
*butyricum* MIYAIRI, *Pediatr Int* 45(1) (2003) 86-90.

[17] J.S. Hu, Y.Y. Huang, J.H. Kuang, J.J. Yu, Q.Y. Zhou, D.M. Liu, *Streptococcus*
*thermophiles* DMST-H2 Promotes Recovery in Mice with Antibiotic-Associated
Diarrhea, *Microorganisms* 8(11) (2020).

[18] J. Pan, G. Gong, Q. Wang, J. Shang, Y. He, C. Catania, D. Birnbaum, Y. Li, Z. Jia,
Y. Zhang, N.S. Joshi, J. Guo, A single-cell nanocoating of probiotics for enhanced
amelioration of antibiotic-associated diarrhea, *Nat Commun* 13(1) (2022) 2117.

[19] Y. Ni, Y. Zhao, L. Ma, Z. Wang, L. Ni, L. Hu, Z. Fu, Pharmacological activation
of REV-ERB α improves nonalcoholic steatohepatitis by regulating intestinal
permeability, *Metabolism* 114 (2021) 154409.

[20] M. Zhu, X. Wang, K. Wang, Z. Zhao, Y. Dang, G. Ji, F. Li, W. Zhou,
*Lingguizhugan* decoction improves non-alcoholic steatohepatitis partially by

modulating gut microbiota and correlated metabolites, *Front Cell Infect Microbiol* 13
(2023) 1066053.

[21] R. Sato, M. Tanaka, Intestinal distribution and intraluminal localization of orally
administered *Clostridium butyricum* in rats, *Microbiol Immunol* 41(9) (1997) 665-71.

[22] P. Louis, H.J. Flint, Diversity, metabolism and microbial ecology of
butyrate-producing bacteria from the human large intestine, *FEMS Microbiol Lett*
294(1) (2009) 1-8.

[23] K.M. Keeney, S. Yurist-Doutsch, M.C. Arrieta, B.B. Finlay, Effects of antibiotics
on human microbiota and subsequent disease, *Annu Rev Microbiol* 68 (2014) 217-35.

[24] M. Willmann, M. Vehreschild, L.M. Biehl, W. Vogel, D. Dörfel, A. Hamprecht, H.
Seifert, I.B. Autenrieth, S. Peter, Distinct impact of antibiotics on the gut microbiome
and resistome: a longitudinal multicenter cohort study, *BMC Biol* 17(1) (2019) 76.

[25] B.P. Willing, S.L. Russell, B.B. Finlay, Shifting the balance: antibiotic effects on
host-microbiota mutualism, *Nat Rev Microbiol* 9(4) (2011) 233-43.

[26] M.K. Stoeva, J. Garcia-So, N. Justice, J. Myers, S. Tyagi, M. Nemchek, P.J.
McMurdie, O. Kolterman, J. Eid, Butyrate-producing human gut symbiont,
*Clostridium butyricum*, and its role in health and disease, *Gut Microbes* 13(1) (2021)
1-28.

[27] L. Jia, D. Li, N. Feng, M. Shamoon, Z. Sun, L. Ding, H. Zhang, W. Chen, J. Sun,
Y.Q. Chen, Anti-diabetic Effects of *Clostridium butyricum* CGMCC0313.1 through
Promoting the Growth of Gut Butyrate-producing Bacteria in Type 2 Diabetic Mice,
*Sci Rep* 7(1) (2017) 7046.

[28] L. Jia, K. Shan, L.L. Pan, N. Feng, Z. Lv, Y. Sun, J. Li, C. Wu, H. Zhang, W.
Chen, J. Diana, J. Sun, Y.Q. Chen, *Clostridium butyricum* CGMCC0313.1 Protects
against Autoimmune Diabetes by Modulating Intestinal Immune Homeostasis and
Inducing Pancreatic Regulatory T Cells, *Front Immunol* 8 (2017) 1345.

[29] J. Liu, Y. Fu, H. Zhang, J. Wang, J. Zhu, Y. Wang, Y. Guo, G. Wang, T. Xu, M.
Chu, F. Wang, The hepatoprotective effect of the probiotic *Clostridium butyricum*
against carbon tetrachloride-induced acute liver damage in mice, *Food Funct* 8(11)
(2017) 4042-4052.

[30] M. Long, S. Yang, P. Li, X. Song, J. Pan, J. He, Y. Zhang, R. Wu, Combined Use
of *C. butyricum* Sx-01 and *L. salivarius* C-1-3 Improves Intestinal Health and
Reduces the Amount of Lipids in Serum via Modulation of Gut Microbiota in Mice,
*Nutrients* 10(7) (2018).

[31] L.L. Pan, W. Niu, X. Fang, W. Liang, H. Li, W. Chen, H. Zhang, M. Bhatia, J.
Sun, *Clostridium butyricum* Strains Suppress Experimental Acute Pancreatitis by
Maintaining Intestinal Homeostasis, *Mol Nutr Food Res* 63(13) (2019) e1801419.

[32] M. Hagihara, Y. Kuroki, T. Ariyoshi, S. Higashi, K. Fukuda, R. Yamashita, A.
Matsumoto, T. Mori, K. Mimura, N. Yamaguchi, S. Okada, T. Nonogaki, T. Ogawa, K.
Iwasaki, S. Tomono, N. Asai, Y. Koizumi, K. Oka, Y. Yamagishi, M. Takahashi, H.
Mikamo, *Clostridium butyricum* Modulates the Microbiome to Protect Intestinal
Barrier Function in Mice with Antibiotic-Induced Dysbiosis, *iScience* 23(1) (2020)
100772.

[33] M. Hagihara, R. Yamashita, A. Matsumoto, T. Mori, Y. Kuroki, H. Kudo, K. Oka,
607 M. Takahashi, T. Nonogaki, Y. Yamagishi, H. Mikamo, The impact of *Clostridium*
*butyricum* MIYAIRI 588 on the murine gut microbiome and colonic tissue, *Anaerobe*
54 (2018) 8-18.

[34] D. Chen, D. Jin, S. Huang, J. Wu, M. Xu, T. Liu, W. Dong, X. Liu, S. Wang, W.
Zhong, Y. Liu, R. Jiang, M. Piao, B. Wang, H. Cao, *Clostridium butyricum*, a
butyrate-producing probiotic, inhibits intestinal tumor development through
modulating Wnt signaling and gut microbiota, *Cancer Lett* 469 (2020) 456-467.

[35] M. Liu, W. Xie, X. Wan, T. Deng, *Clostridium butyricum* modulates gut
microbiota and reduces colitis associated colon cancer in mice, *Int Immunopharmacol*
88 (2020) 106862.

[36] E.M. Velazquez, H. Nguyen, K.T. Heasley, C.H. Saechao, L.M. Gil, A.W.L.
Rogers, B.M. Miller, M.R. Rolston, C.A. Lopez, Y. Litvak, M.J. Liou, F. Faber, D.N.
Bronner, C.R. Tiffany, M.X. Byndloss, A.J. Byndloss, A.J. Bäumlner, Endogenous
Enterobacteriaceae underlie variation in susceptibility to *Salmonella* infection, *Nat*
*Microbiol* 4(6) (2019) 1057-1064.

[37] S.J. Robertson, P. Lemire, H. Maughan, A. Goethel, W. Turpin, L. Bedrani, D.S.

Guttman, K. Croitoru, S.E. Girardin, D.J. Philpott, Comparison of Co-housing and
Littermate Methods for Microbiota Standardization in Mouse Models, *Cell Rep* 27(6)
(2019) 1910-1919.e2.

[38] A.C. Ericsson, J.W. Davis, W. Spollen, N. Bivens, S. Givan, C.E. Hagan, M.
McIntosh, C.L. Franklin, Effects of vendor and genetic background on the
composition of the fecal microbiota of inbred mice, *PLoS One* 10(2) (2015)
e0116704.

[39] E.F. Tigchelaar, M.J. Bonder, S.A. Jankipersadsing, J. Fu, C. Wijmenga, A.
Zhernakova, Gut microbiota composition associated with stool consistency, *Gut* 65(3)
(2016) 540-2.

[40] J. König, J. Wells, P.D. Cani, C.L. García-Ródenas, T. MacDonald, A. Mercenier,
634 J. Whyte, F. Troost, R.J. Brummer, Human Intestinal Barrier Function in Health and
635 Disease, *Clin Transl Gastroenterol* 7(10) (2016) e196.

[41] M.V. Tulstrup, E.G. Christensen, V. Carvalho, C. Linnige, S. Ahrné, O. Højberg,
637 T.R. Licht, M.I. Bahl, Antibiotic Treatment Affects Intestinal Permeability and Gut
Microbial Composition in Wistar Rats Dependent on Antibiotic Class, *PLoS One*
10(12) (2015) e0144854.

[42] A. Pellegrino, G. Coppola, F. Santopaolo, A. Gasbarrini, F.R. Ponziani, Role of
*Akkermansia* in Human Diseases: From Causation to Therapeutic Properties,
*Nutrients* 15(8) (2023).

[43] Y. Ji, Y. Yin, Z. Li, W. Zhang, Gut Microbiota-Derived Components and
Metabolites in the Progression of Non-Alcoholic Fatty Liver Disease (NAFLD),
*Nutrients* 11(8) (2019).

[44] A.K. Duttaroy, Role of Gut Microbiota and Their Metabolites on Atherosclerosis,
Hypertension and Human Blood Platelet Function: A Review, *Nutrients* 13(1) (2021).

[45] X. Jia, W. Xu, L. Zhang, X. Li, R. Wang, S. Wu, Impact of Gut Microbiota and
Microbiota-Related Metabolites on Hyperlipidemia, *Front Cell Infect Microbiol* 11
(2021) 634780.

[46] A.E. Pérez-Cobas, M.J. Gosalbes, A. Friedrichs, H. Knecht, A. Artacho, K.
Eismann, W. Otto, D. Rojo, R. Bargiela, M. von Bergen, S.C. Neulinger, C. Däumer,

F.A. Heinsen, A. Latorre, C. Barbas, J. Seifert, V.M. dos Santos, S.J. Ott, M. Ferrer, A.
Moya, Gut microbiota disturbance during antibiotic therapy: a multi-omic approach,
Gut 62(11) (2013) 1591-601.

[47] P. Louis, G.L. Hold, H.J. Flint, The gut microbiota, bacterial metabolites and
colorectal cancer, Nat Rev Microbiol 12(10) (2014) 661-72.

[48] C.M. van der Beek, C.H.C. Dejong, F.J. Troost, A.A.M. Masclee, K. Lenaerts,
Role of short-chain fatty acids in colonic inflammation, carcinogenesis, and mucosal
protection and healing, Nutr Rev 75(4) (2017) 286-305.

[49] H.J. Binder, Role of colonic short-chain fatty acid transport in diarrhea, Annu
Rev Physiol 72 (2010) 297-313.

[50] M. Wullt, M.L. Johansson Hagslätt, I. Odenholt, Lactobacillus plantarum 299v
enhances the concentrations of fecal short-chain fatty acids in patients with recurrent
clostridium difficile-associated diarrhea, Dig Dis Sci 52(9) (2007) 2082-6.

[51] G. Cresci, L.E. Nagy, V. Ganapathy, Lactobacillus GG and tributyrin
supplementation reduce antibiotic-induced intestinal injury, JPEN J Parenter Enteral
Nutr 37(6) (2013) 763-74.

[52] X. Song, X. Sun, S.F. Oh, M. Wu, Y. Zhang, W. Zheng, N. Geva-Zatorsky, R.
Jupp, D. Mathis, C. Benoist, D.L. Kasper, Microbial bile acid metabolites modulate
gut ROR γ (+) regulatory T cell homeostasis, Nature 577(7790) (2020) 410-415.

[53] W. Jia, G. Xie, W. Jia, Bile acid-microbiota crosstalk in gastrointestinal
inflammation and carcinogenesis, Nat Rev Gastroenterol Hepatol 15(2) (2018)
111-128.

[54] J.A. Winston, C.M. Theriot, Diversification of host bile acids by members of the
gut microbiota, Gut Microbes 11(2) (2020) 158-171.

**Figure legends**

**Figure 1. *C. butyricum* RH2 ameliorated antibiotic-associated diarrhea (AAD)**
**symptoms in juvenile mice.**

[revised manuscript text omitted]

734

From my perspective of view, in this manuscript:

Line36 : Repetition: the phrases "young mice" and "juvenile mice" appeared twice; the term can be unified.

Line 45: The flow between the sentence describing the results and the concluding sentence can be improved (the last sentence appears more appended than integrated).

Line 151 : The reference (housekeeping) gene used for normalizing gene expression (such as GAPDH or β -actin) was not specified.

Line 153: The reaction volume, RNA concentration, and thermal cycling conditions for qPCR were not specified, and these are often required.

Line 221: For better flow: the transition from describing the experiment to presenting the results (weight loss, lethargy) was abrupt; it would be preferable to add a linking sentence.

Line 369: Given that SCFAs and bile acids play central roles in gut homeostasis, how can we determine whether the observed therapeutic effect of *C. butyricum* RH2 is primarily driven by SCFA production, bile acid modulation, or synergistic interactions between both pathways?

Line 434: Given that *C. butyricum* is a transient rather than colonizing species, to what extent can its long-term ecological and therapeutic effects be sustained after discontinuation of supplementation, and how might this impact its clinical applicability in AAD treatment?"

***Clostridium butyricum* RH2 ameliorates Diarrhea in juvenile mice under**
**continuous antibiotics exposure by modulating gut microbiota and metabolome**

Yufeng Zhao¹, Yunlong Shang¹, Shen Cheng¹, Tianyue Guan¹, Fengyu Guo¹, Yu
Wang^{1,*}

¹Hangzhou Grand Biologic Pharmaceutical Inc., Hangzhou, 310000, China

*Corresponding author.

Yu Wang, PhD

Hangzhou Grand Biologic Pharmaceutical Inc., Hangzhou, 310000, China

Tel: 86-571-2802-0125

Fax: 86-571-2802-0125

E-mail: wangyu@hzydsw.cn

Word count: 5247

**Abbreviations**

Antibiotic-associated diarrhea: AAD; *Clostridium butyricum*: *C. butyricum*; Claudin-1:
Cldn1; Claudin-5: Cldn5; Occludin: Ocln; Tight junction protein 1: Tjp1; Cadherin 1:
Cdh1; Interleukin 6: Il6; Tumor necrosis factor α : Tnf α ; Interleukin 10: Il10;
Interleukin 1 beta: Il1 β ; Interleukin 4: Il4; Short-chain fatty acids: SCFAs

**Abstract:**

Antibiotic-associated diarrhea (AAD) is a self-limiting disorder triggered by
antibiotic therapy in pediatric populations. Although multiple probiotics are clinically
employed for AAD management, the therapeutic efficacy of *Clostridium butyricum*
(*C. butyricum*) in pediatric AAD and its underlying mechanisms remain poorly
characterized. This study aimed to establish a juvenile mice model of AAD and
investigate the therapeutic potential of oral *C. butyricum* administration in young mice
subjected to continuous antibiotics exposure. We systematically assessed pathological
changes in colonic tissue, colitis severity, intestinal epithelial barrier integrity, fecal
metabolomic profiles, and gut microbiota diversity. Our analysis demonstrates that *C.*
*butyricum* ameliorates intestinal inflammation, enhances barrier function by
modulating the gut microbiota and its metabolites, and significantly alleviates
diarrhea symptoms in juvenile AAD mice. Furthermore, this study reveals that *C.*
*butyricum* can tolerate continuous antibiotics exposure within the gastrointestinal tract
of young mice, providing a scientific rationale for its co-administration with
antibiotics.

**Keywords:** antibiotic-associated diarrhea; gut barrier function; *Clostridium butyricum*
RH2; gut microbiota; metabolomics.

**Introduction**

As the most frequent complication of antimicrobial treatment,
antibiotic-associated diarrhea (AAD) is defined as unexplained diarrheal episodes
temporally linked to antibiotic use. All antibiotic classes demonstrate diarrheagenic
potential, particularly broad-spectrum agents including penicillins, cephalosporins,
and clindamycin[1]. The clinical manifestations depend on encompassing antibiotic
class, dosage regimen, treatment duration, administration route, and combination
therapies[2]. While AAD affects all age demographics, pediatric populations exhibit
heightened susceptibility, with reported prevalence rates of 20-35% in children[3].
Epidemiological surveillance data indicate substantial variability across care settings:
US pediatric cohorts demonstrate 6% incidence in ambulatory care versus 80% in
hospitalized patients[3]. Chinese clinical studies, currently limited to inpatient settings,
report incidence rates ranging from 16.8% to 70.6%[4]. Current management remains
restricted to antibiotic discontinuation or substitution, with no targeted therapies
available. Pathophysiological mechanisms involve antibiotic-induced dysbiosis
characterized by gastrointestinal microbiota disruption, opportunistic pathogen
proliferation, and metabolic dysfunction.

Probiotics are defined as live microorganisms that confer health benefits to the
host when administered in adequate doses[5]. Substantial evidence from preclinical
and clinical studies has established that antibiotic-induced gut microbiota dysbiosis
plays a fundamental role in AAD pathogenesis. This mechanistic understanding
supports probiotics as viable therapeutic agents for microbiota restoration. Multiple
randomized controlled trials have demonstrated the efficacy of specific microbial
strains in AAD prevention, including
*Bifidobacterium* spp., *Clostridium* spp., *Lactobacillus* spp., and the fungal
species *Saccharomyces boulardii* [6-13].

*Clostridium butyricum* (*C. butyricum*), a strictly anaerobic spore-forming
bacillus, has been extensively utilized as a probiotic in Asian countries for decades.
This microorganism demonstrates ubiquitous distribution, inhabiting soil, dairy
products, vegetable matter, and the human gastrointestinal tract. As an early gut

colonizer, it is detected in 10-20% of adult populations[14]. Notably, *C. butyricum*
encompasses multiple strains exhibiting diverse health-promoting mechanisms
through host-microbe interactions[15]. Emerging evidence positions this species as a
multifunctional therapeutic agent, demonstrating efficacy in gastrointestinal,
neurological, and metabolic disorders, alongside potential anticancer properties[16].
However, the potential mechanism by which *C. butyricum* RH2 alleviates diarrhea
symptoms induced by multiple antibiotics in juvenile mice has not been explored.

In this study, we employed *C. butyricum* RH2 strain to investigate its therapeutic
efficacy and mechanistic underpinnings in a juvenile murine model of
antibiotic-associated diarrhea (AAD). Given the heightened clinical vulnerability of
pediatric populations to AAD complications, we established a developmentally
relevant mouse model using 4-week-old C57BL/6J mice. The model was induced
through sequential oral administration of an antibiotic cocktail (ampicillin,
streptomycin, and clindamycin), simulating pediatric clinical conditions. Our findings
demonstrate that *C. butyricum* RH2 exhibits significant potential as a probiotic
adjuvant for maintaining intestinal homeostasis in antibiotic-dependent pediatric AAD
management.

**Materials and Methods**

**Reagents and Probiotics Strain**

The juvenile AAD murine model was established through daily gavage
administration of an antibiotic cocktail containing clindamycin (Jiudian, Hunan,
China), ampicillin (United Laboratories, Hong Kong, China), and streptomycin
(Lukang, Shandong, China) dissolved in normal saline. Antibiotic cocktail
administration was performed daily at 10:00.

*C. butyricum* RH2 lyophilized powder (Grand Biologic Pharmaceutical; Batch
J202304072) was reconstituted in sterile 0.9% NaCl solution to achieve target
concentrations. A minimum 4-hour interval was maintained between probiotic and
antibiotic administrations to prevent pharmacological interference.

**Animal Experimental Design**

Sixty specific pathogen-free (SPF) male C57BL/6J mice (3-week-old) were
obtained from Shanghai SLAC Laboratory Animal Co., Ltd. (Shanghai, China). The
animals were maintained under controlled environmental conditions (temperature: 22
\pm 2°C; humidity: 40 \pm 5%) with a 12-hour light/dark cycle, and provided ad libitum
access to food and water. All experimental protocols complied with the National
Institutes of Health Guide for the Care and Use of Laboratory Animals and were
approved by the Institutional Animal Care and Use Committee of Hangzhou Qingda
Kerui Biotechnology Co., Ltd. (Approval No.: QDAE20230625001).

As illustrated in Figure 1A, following a 7-day acclimatization period, mice were
randomly allocated into six experimental groups: normal control (NC),
antibiotic-associated diarrhea model (M), antibiotic withdrawal control (AW), and
three *C. butyricum* RH2 treatment groups receiving low (LD: 1.05×10^5 CFU daily),
medium (MD: 1.05×10^7 CFU daily), or high doses (HD: 1.05×10^9 CFU daily).
Based on established protocols demonstrating successful AAD model induction
through 3-day antibiotic administration [17], therapeutic interventions commenced
immediately post-antibiotic treatment and continued for 14 days. To prevent
spontaneous recovery reported in untreated AAD models[17], antibiotic
administration was maintained throughout the treatment phase. The AW group served
as a self-recovery control, receiving antibiotics for 3 days followed by 14 days of
saline treatment. Fecal samples were collected from all groups for subsequent
analyses.

**Diarrhea Assessment**

Diarrhea severity was quantitatively assessed through fecal consistency scoring
and fecal moisture content analysis following established methodology[18].
Antibiotic-associated diarrhea status was classified based on the following visual
grading scale: (1) Alert behavioral status with formed, the feces are elliptical in shape,
with a hard texture and a brownish color, score = 1; (2) General mental state, the feces
are sausage shaped, with a smooth surface and a yellowish color, score = 2; and (3)
Bad mental state, the feces are in the shape of sausages or paste, with a wet and soft
texture, and a yellowish color, score =3. Fresh fecal samples were collected over a

standardized 1-hour period using metabolic cages. Samples were immediately
weighed (wet weight), desiccated at 60°C to constant mass, and reweighed (dry
weight). Fecal moisture content was calculated using:

$$144 \quad \text{Fecal Water Content (\%)} = [(\text{Wet Weight} - \text{Dry Weight})/\text{Wet Weight}] \times 100$$

**RNA extraction and RT-PCR**

Total RNA was isolated from tissue specimens using TRIzol™ Reagent (Thermo
Fisher Scientific, Waltham, MA, USA) following the manufacturer's protocol.
First-strand cDNA synthesis was performed with the HiScript III Reverse
Transcriptase Kit (Vazyme Biotech Co., Ltd., Nanjing, China). Quantitative real-time
polymerase chain reaction (qPCR) analyses were conducted on a QuantStudio™ 7500
Fast Real-Time PCR System (Applied Biosystems, Foster City, CA, USA) using
ChamQ Universal SYBR qPCR Master Mix (Vazyme Biotech Co., Ltd.), as
previously described[19]. The primer sequences are shown in Supplementary Table 1.

**Histological examination**

For histological analysis, colon tissue samples were fixed in 4%
paraformaldehyde and embedded in paraffin. Hematoxylin and eosin (H&E) staining
was performed according to established protocols[19].

**16S rRNA gene amplicon Sequencing**

Extract bacterial DNA from mice feces were obtained using OMEGA Soil DNA
Kit (M5635-02; OMEGA Bio-Tek, Norcross, GA, USA). NanoDrop NC2000
spectrophotometer (Thermo Fisher Scientific, Waltham, MA, USA) followed by
agarose gel electrophoresis were used to measure the quality of extracted DNA. PCR
amplification was performed on the V3-V4 region of the bacterial 16S rRNA gene
using forward primer 338F (5'-ACTCCTACGGGAGGCAGCA-3') and reverse
primer 806R (5' -GGACTACHVGGGTWTCTAAT-3'). Vazyme VAHTSTM DNA
Clean Beads (Vazyme, Nanjing, China) was used to purify PCR amplicons, which
quantifying with QuantiT PicoGreen dsDNA assay kit (Invitrogen, Carlsbad, CA,
USA). Purified amplicons were merged in equal quantities and paired for sequencing
using Illumina NovaSeq platform and NovaSeq6000 SP kit (500 cycles). The

sequence data were processed using QIIME2 according to previously described
methods[20].

**Metabolomics analysis**

Metabolites in fecal samples were quantified using an ultra-performance liquid
chromatography-tandem mass spectrometry (UPLC-MS/MS) system (ACQUITY
UPLC-Xevo TQ-S; Waters Corporation, Milford, MA, USA) with the Q300
Metabolite Array Kit (Metabo-Profile Biotechnology, Shanghai, China). Raw data
files generated by UPLC -MS/MS were processed using iMAP platform (v1.0;
Metabo-Profile, China). Multivariate statistical analysis (ANOVA or Kruskal-Wallis
test, selected based on data normality and variance homogeneity) was applied to
perform principal component analysis (PCA) and identify differential metabolites
between groups. Differential metabolites were defined by a significance threshold of
$P < 0.05$. Shared metabolites among pairwise comparisons (NC vs. M, M vs. AW, and
M vs. HD) were visualized using Venn diagrams, while hierarchical clustering
analysis illustrated metabolite expression patterns across groups. Metabolite
enrichment analysis for *C. butyricum* RH2-reversed metabolic alterations was
conducted through the MetaboAnalyst platform.

**Statistics**

All data were expressed as the mean \pm standard error of the mean (SEM).
One-way ANOVA among groups followed by Student's t-test between groups were
performed to analyze the differences using the Prism 9.0 program (GraphPad
Software, San Diego, Canada). Spearman correlation analyses between gut microbiota
composition, fecal metabolites, and antibiotic-associated diarrhea (AAD) indices were
performed using SPSS version 24 (IBM Corp., Armonk, NY, USA). Statistical
significance was defined as $p < 0.05$.

**Data Availability Statement**

Sequencing datasets have been deposited to the NCBI Sequence Read Archive
under BioProject accession numbers PRJNA1157009.

**Disclosure of potential conflicts of interest**

The authors disclose the following potential conflicts of interest: Y.Z., Y.S., S.C.,

[revised manuscript text omitted]

Antibiotic-associated diarrhea (AAD) is widely recognized as a common adverse
reaction to antibiotic therapy, particularly in pediatric populations. Current pediatric
AAD management guidelines recommend antibiotic withdrawal as a primary
therapeutic strategy, provided that discontinuation does not compromise treatment of

the underlying infection. To investigate the pathophysiology of pediatric AAD, we
developed a juvenile murine model using C57BL/6J mice administered with
broad-spectrum antibiotics, which reliably induced diarrheal symptoms characteristic
of clinical AAD. *C. butyricum* RH2, a Gram-positive, obligate anaerobic bacterium
within Clostridium cluster I, demonstrates significant butyrogenic capacity[16, 21].
Butyrate-producing microbiota are increasingly recognized for their
anti-inflammatory properties through epithelial barrier reinforcement and immune
modulation[22]. Our investigation revealed that supplementing *C. butyricum* could
effectively alleviate AAD-related symptoms, as evidenced by water in stool and
diarrhea score diminished (Figure 1), colon inflammation ameliorated (Figure 2).
Furthermore, we confirmed that *C. butyricum* RH2 can modulate the gut microbiota
and metabolites, protected gut barrier function and improved gut microbiota
homeostasis, which also helps to improve the symptoms of pathological diarrhea.
Notably, therapeutic outcomes including diarrhea severity (Figure 1C), colonic
shortening (Figure 2B), and intestinal barrier parameters (Figure 3) demonstrated
comparable efficacy between continuous antibiotic therapy with *C. butyricum* RH2
supplementation and complete antibiotic withdrawal. This finding suggests that
concurrent administration of *C. butyricum* RH2 with essential antibiotic therapy may
prevent AAD development in pediatric patients requiring prolonged antimicrobial
treatment.

The gastrointestinal tract serves as the principal colonization site for human
microbiota, harboring 1000 types of approximately 10^{13} - 10^{14} bacteria residing in the
normal human gut[23]. Various bacteria in the gut microbiota constrain each other
through occupying effects, nutritional competition, metabolic mutualism, production
of organic acids and antibacterial substances, and immune synergy mechanisms,
forming a balance between the microbiota in a certain composition and proportion[24].
Antibiotics cause significant disruptions to the normal composition and functional
attributes of the gut microbiome, such alterations can persist well after the cessation
of antibiotic administration[25]. Similarly, the withdraw of antibiotics for 14 days did

not restore the fecal microbial diversity and richness of AAD juvenile mice in the
present study (Supplementary Figure 2). Dysbiosis of the microbiota induced by
antibiotics represent that inhibition of sensitive microbiota in the intestine, leading to
excessive proliferation of non-sensitive microbiota, decreased diversity of gut
microbiota, and profound changes in the composition and proportion of
microbiota[25]. Thus, the gut microbiota might play a critical role in the pathogenesis
and progression of AAD.

The 16S rRNA gene sequencing approach, involving PCR amplification of
hypervariable regions (V3-V5) from fecal samples followed by Illumina deep
sequencing, represents the predominant methodology for investigating *C. butyricum*'s
impact on gut microbiota composition[26]. Current research presents conflicting
findings regarding microbial community responses to *C. butyricum* supplementation.
While multiple studies report significant alterations in α -diversity indices and
β -diversity patterns[27-32], other investigations, including our current work
(Supplemental Figure 2), demonstrate no statistically significant changes in these
microbial community metrics [33-35]. These discrepancies may stem from
methodological variations in 16S rRNA sequencing protocols and multiple
confounding factors affecting gut microbiota profiles, including host genotype,
environmental conditions (housing/litter), dietary composition, and vendor-specific
microbiome signatures [36-38].

As a transient commensal bacterium, *C. butyricum* exhibits unique ecological
characteristics in the gastrointestinal tract. This non-colonizing species demonstrates
synergistic relationships with beneficial gut microbiota through trophic interactions
while simultaneously suppressing pathogenic and putrefactive bacterial populations
via competitive exclusion and metabolic inhibition. These ecological patterns are
corroborated by our 16S rRNA sequencing analyses, revealing that high-dose *C.*
*butyricum* RH2 administration significantly elevated the relative abundance of
beneficial genera (*Ruminococcus*, *Oscillospira*, *Bacteroides*, and *Akkermansia*) while
suppressing pathogenic taxa (*Acinetobacter*, *Staphylococcus*, and *Pseudomonas*). This
microbial modulation aligns with established clinical correlations - a European

population study demonstrated that *Oscillospira* abundance positively associates with
stool firmness and negatively correlates with diarrheal status[39]. Our findings
substantiate this relationship through a significant inverse correlation between fecal
water content and *Oscillospira* abundance (Figure 6B).

Accumulating evidence underscores the critical role of intestinal epithelial
barrier integrity in the pathogenesis of various intestinal and systemic disorders[40].
Studies have shown that the use of antibiotics can lead to barrier dysfunction or
“intestinal leakage”[41]. A key characteristic of *A. muciniphila* lies in its capacity to
degrade intestinal mucin glycoproteins through fermentation processes, generating
sulfate metabolites along with short-chain fatty acids including acetate and
propionate. This mucolytic activity facilitates mucosal layer renewal and structural
reinforcement, thereby improving barrier function and limiting microbial
translocation[42]. In this study, we found a thinner, penetrable mucus layer and a
rapid reduction in the expression of *Muc2*, *Cdh1*, *Ocln*, *Cldn1*, *Cldn5* and *Tjp1* in the
AAD mice, while *C. butyricum* RH2 treatment alleviated antibiotic-induced barrier
dysfunction (Figure 3). These results suggested that the reduced severity of diarrhea
in the supplementing the *C. butyricum* RH2 group may be attributed to the protection
of gut barrier integrity (Figure 3). Consistently, the feces of AAD mice supplemented
with *C. butyricum* RH2 showed a significant increase in the genus *Akkermansia* in our
study (Figure 4F), and the correlation analysis also suggested that *Akkermansia* was
positively linked to tight junction protein gene expression but negatively associated
with diarrhea-related index and proinflammatory gene expression(Figure 6A),
indicating that *C. butyricum* RH2 improved the intestinal barrier function and
alleviated diarrhea symptoms in AAD mice by upregulating the genus *Akkermansia*.
Taken together, our study confirmed that probiotic *C. butyricum* RH2 was able to
improve the gastrointestinal symptoms of AAD and induce *Oscillospira* and
*Akkermansia* to gradually become the dominant species in the gut.

Mounting evidence indicates that gut microbiota-derived metabolites play a
pivotal role in modulating the pathogenesis of metabolic disorders[43-45].
Antibiotic-induced gut microbiota remodeling has been shown to significantly alter

intestinal metabolomic profiles[46]. AAD is typically characterized by decreased
intestinal short chain fatty acids (SCFAs) concentrations, accumulation of luminal
carbohydrates and colonic bile acids, altered water absorption, and ultimately
diarrhea[47]. Our metabolomic analysis identified several fecal metabolites including
short-chain fatty acids (SCFAs), bile acids (BAs), amino acids (AAs), etc. Short chain
fatty acids(SCFAs) are rapidly absorbed by the colon and stimulate Na-dependent
fluid absorption via a cyclic AMP-independent process with Na-H, SCFA-HCO₃, and
Cl-SCFAs exchanges, which means the reduction levels of SCFAs might lead to AAD
because of the NaCl and water absorption was damaged [48, 49]. More and more
probiotics are reported that can directly promote intestinal SCFAs by producing
organic acids, lactic acid, and acetic acid, or providing a more suitable environment
for SCFAs producing bacteria[50, 51]. Our research also confirms that supplementing
*C. butyricum* RH2 can effectively elevates fecal SCFA concentrations (Figure 5).

Bile acids (BAs) are the metabolic products of dietary substrates and play a vital
role in metabolism and immunological regulation[52, 53]. Approximately 95% of
luminal bile acids are reabsorbed in the distal ileum in healthy individuals, the
remaining amounts are modified by intestinal bacteria and then are either excreted or
passively absorbed[54]. Excessive secretion of bile acids or insufficient absorption of
bile acids lead to surplus bile acids present in the intestine, which increasing the
permeability of the intestinal mucosa and affecting the absorption of water and
sodium in the intestine, ultimately resulting in a large amount of intestinal fluid and
causing diarrhea[52, 53]. Our findings corroborate this mechanism, demonstrating
significant positive correlations between two specific bile acids—taurodeoxycholic
acid (TDCA) and tauroursodeoxycholic acid (TUDCA)—and key AAD clinical
markers, including fecal water content and diarrhea severity scores (Figure 6A).

Overall, we successfully established a novel antibiotic-associated diarrhea (AAD)
model in 4-week-old juvenile mice using a triple-antibiotic combination
(streptomycin, ampicillin, and clindamycin). Under continuous antibiotic exposure, *C.*
*butyricum* RH2 conferred intestinal protection by maintaining gut homeostasis
through modulation of microbiota composition and metabolites, enhancement of

intestinal epithelial barrier integrity, and suppression of colitis. Furthermore, this
study provides nonclinical evidence supporting the potential application of probiotic
*C. butyricum* to alleviate diarrhea in pediatric patients requiring continuous antibiotic
therapy.

**References**

- [1] L.V. McFarland, Antibiotic-associated diarrhea: epidemiology, trends and
treatment, *Future Microbiol* 3(5) (2008) 563-78.
- [2] C.L.R. Abad, N. Safdar, A Review of *Clostridioides difficile* Infection and
Antibiotic-Associated Diarrhea, *Gastroenterol Clin North Am* 50(2) (2021) 323-340.
- [3] L.V. McFarland, M. Ozen, E.C. Dinleyici, S. Goh, Comparison of pediatric and
adult antibiotic-associated diarrhea and *Clostridium difficile* infections, *World J*
*Gastroenterol* 22(11) (2016) 3078-104.
- [4] Q. Yang, Z. Hu, Y. Lei, X. Li, C. Xu, J. Zhang, H. Liu, X. Du, Overview of
systematic reviews of probiotics in the prevention and treatment of
antibiotic-associated diarrhea in children, *Front Pharmacol* 14 (2023) 1153070.
- [5] C. Hill, F. Guarner, G. Reid, G.R. Gibson, D.J. Merenstein, B. Pot, L. Morelli, R.B.
Canani, H.J. Flint, S. Salminen, P.C. Calder, M.E. Sanders, Expert consensus
document. The International Scientific Association for Probiotics and Prebiotics
consensus statement on the scope and appropriate use of the term probiotic, *Nat Rev*
*Gastroenterol Hepatol* 11(8) (2014) 506-14.
- [6] M. Storr, A. Stengel, Systematic review: clinical evidence of probiotics in the
prevention of antibiotic-associated diarrhoea, *MMW Fortschr Med* 163(Suppl 4)
(2021) 19-26.
- [7] Q. Guo, J.Z. Goldenberg, C. Humphrey, R. El Dib, B.C. Johnston, Probiotics for
the prevention of pediatric antibiotic-associated diarrhea, *Cochrane Database Syst*
*Rev* 4(4) (2019) Cd004827.
- [8] H.B. Xu, R.H. Jiang, H.B. Sheng, Meta-analysis of the effects of *Bifidobacterium*
preparations for the prevention and treatment of pediatric antibiotic-associated
diarrhea in China, *Complement Ther Med* 33 (2017) 105-113.
- [9] H. Szajewska, M. Kołodziej, Systematic review with meta-analysis:
*Saccharomyces boulardii* in the prevention of antibiotic-associated diarrhoea, *Aliment*
*Pharmacol Ther* 42(7) (2015) 793-801.
- [10] H. Szajewska, M. Kołodziej, Systematic review with meta-analysis:
*Lactobacillus rhamnosus* GG in the prevention of antibiotic-associated diarrhoea in

children and adults, *Aliment Pharmacol Ther* 42(10) (2015) 1149-57.

[11] H. Szajewska, M. Ruszczyński, A. Radzikowski, Probiotics in the prevention of
antibiotic-associated diarrhea in children: a meta-analysis of randomized controlled
trials, *J Pediatr* 149(3) (2006) 367-372.

[12] H. Szajewska, R.B. Canani, A. Guarino, I. Hojsak, F. Indrio, S. Kolacek, R. Orel,
R. Shamir, Y. Vandenplas, J.B. van Goudoever, Z. Weizman, Probiotics for the
Prevention of Antibiotic-Associated Diarrhea in Children, *J Pediatr Gastroenterol*
*Nutr* 62(3) (2016) 495-506.

[13] B.C. Johnston, J.Z. Goldenberg, P.O. Vandvik, X. Sun, G.H. Guyatt, Probiotics
for the prevention of pediatric antibiotic-associated diarrhea, *Cochrane Database Syst*
*Rev* (11) (2011) Cd004827.

[14] K.C. Mountzouris, A.L. McCartney, G.R. Gibson, Intestinal microflora of human
infants and current trends for its nutritional modulation, *Br J Nutr* 87(5) (2002)
405-20.

[15] N. Cassir, S. Benamar, B. La Scola, *Clostridium butyricum*: from beneficial to a
new emerging pathogen, *Clin Microbiol Infect* 22(1) (2016) 37-45.

[16] H. Seki, M. Shiohara, T. Matsumura, N. Miyagawa, M. Tanaka, A. Komiyama, S.
Kurata, Prevention of antibiotic-associated diarrhea in children by *Clostridium*
*butyricum* MIYAIRI, *Pediatr Int* 45(1) (2003) 86-90.

[17] J.S. Hu, Y.Y. Huang, J.H. Kuang, J.J. Yu, Q.Y. Zhou, D.M. Liu, *Streptococcus*
*thermophiles* DMST-H2 Promotes Recovery in Mice with Antibiotic-Associated
Diarrhea, *Microorganisms* 8(11) (2020).

[18] J. Pan, G. Gong, Q. Wang, J. Shang, Y. He, C. Catania, D. Birnbaum, Y. Li, Z. Jia,
Y. Zhang, N.S. Joshi, J. Guo, A single-cell nanocoating of probiotics for enhanced
amelioration of antibiotic-associated diarrhea, *Nat Commun* 13(1) (2022) 2117.

[19] Y. Ni, Y. Zhao, L. Ma, Z. Wang, L. Ni, L. Hu, Z. Fu, Pharmacological activation
of REV-ERB α improves nonalcoholic steatohepatitis by regulating intestinal
permeability, *Metabolism* 114 (2021) 154409.

[20] M. Zhu, X. Wang, K. Wang, Z. Zhao, Y. Dang, G. Ji, F. Li, W. Zhou,
*Lingguizhugan* decoction improves non-alcoholic steatohepatitis partially by

modulating gut microbiota and correlated metabolites, *Front Cell Infect Microbiol* 13
(2023) 1066053.

[21] R. Sato, M. Tanaka, Intestinal distribution and intraluminal localization of orally
administered *Clostridium butyricum* in rats, *Microbiol Immunol* 41(9) (1997) 665-71.

[22] P. Louis, H.J. Flint, Diversity, metabolism and microbial ecology of
butyrate-producing bacteria from the human large intestine, *FEMS Microbiol Lett*
294(1) (2009) 1-8.

[23] K.M. Keeney, S. Yurist-Doutsch, M.C. Arrieta, B.B. Finlay, Effects of antibiotics
on human microbiota and subsequent disease, *Annu Rev Microbiol* 68 (2014) 217-35.

[24] M. Willmann, M. Vehreschild, L.M. Biehl, W. Vogel, D. Dörfel, A. Hamprecht, H.
Seifert, I.B. Autenrieth, S. Peter, Distinct impact of antibiotics on the gut microbiome
and resistome: a longitudinal multicenter cohort study, *BMC Biol* 17(1) (2019) 76.

[25] B.P. Willing, S.L. Russell, B.B. Finlay, Shifting the balance: antibiotic effects on
host-microbiota mutualism, *Nat Rev Microbiol* 9(4) (2011) 233-43.

[26] M.K. Stoeva, J. Garcia-So, N. Justice, J. Myers, S. Tyagi, M. Nemchek, P.J.
McMurdie, O. Kolterman, J. Eid, Butyrate-producing human gut symbiont,
*Clostridium butyricum*, and its role in health and disease, *Gut Microbes* 13(1) (2021)
1-28.

[27] L. Jia, D. Li, N. Feng, M. Shamoon, Z. Sun, L. Ding, H. Zhang, W. Chen, J. Sun,
Y.Q. Chen, Anti-diabetic Effects of *Clostridium butyricum* CGMCC0313.1 through
Promoting the Growth of Gut Butyrate-producing Bacteria in Type 2 Diabetic Mice,
*Sci Rep* 7(1) (2017) 7046.

[28] L. Jia, K. Shan, L.L. Pan, N. Feng, Z. Lv, Y. Sun, J. Li, C. Wu, H. Zhang, W.
Chen, J. Diana, J. Sun, Y.Q. Chen, *Clostridium butyricum* CGMCC0313.1 Protects
against Autoimmune Diabetes by Modulating Intestinal Immune Homeostasis and
Inducing Pancreatic Regulatory T Cells, *Front Immunol* 8 (2017) 1345.

[29] J. Liu, Y. Fu, H. Zhang, J. Wang, J. Zhu, Y. Wang, Y. Guo, G. Wang, T. Xu, M.
Chu, F. Wang, The hepatoprotective effect of the probiotic *Clostridium butyricum*
against carbon tetrachloride-induced acute liver damage in mice, *Food Funct* 8(11)
(2017) 4042-4052.

[30] M. Long, S. Yang, P. Li, X. Song, J. Pan, J. He, Y. Zhang, R. Wu, Combined Use
of *C. butyricum* Sx-01 and *L. salivarius* C-1-3 Improves Intestinal Health and
Reduces the Amount of Lipids in Serum via Modulation of Gut Microbiota in Mice,
*Nutrients* 10(7) (2018).

[31] L.L. Pan, W. Niu, X. Fang, W. Liang, H. Li, W. Chen, H. Zhang, M. Bhatia, J.
Sun, *Clostridium butyricum* Strains Suppress Experimental Acute Pancreatitis by
Maintaining Intestinal Homeostasis, *Mol Nutr Food Res* 63(13) (2019) e1801419.

[32] M. Hagihara, Y. Kuroki, T. Ariyoshi, S. Higashi, K. Fukuda, R. Yamashita, A.
Matsumoto, T. Mori, K. Mimura, N. Yamaguchi, S. Okada, T. Nonogaki, T. Ogawa, K.
Iwasaki, S. Tomono, N. Asai, Y. Koizumi, K. Oka, Y. Yamagishi, M. Takahashi, H.
Mikamo, *Clostridium butyricum* Modulates the Microbiome to Protect Intestinal
Barrier Function in Mice with Antibiotic-Induced Dysbiosis, *iScience* 23(1) (2020)
100772.

[33] M. Hagihara, R. Yamashita, A. Matsumoto, T. Mori, Y. Kuroki, H. Kudo, K. Oka,
607 M. Takahashi, T. Nonogaki, Y. Yamagishi, H. Mikamo, The impact of *Clostridium*
*butyricum* MIYAIRI 588 on the murine gut microbiome and colonic tissue, *Anaerobe*
54 (2018) 8-18.

[34] D. Chen, D. Jin, S. Huang, J. Wu, M. Xu, T. Liu, W. Dong, X. Liu, S. Wang, W.
Zhong, Y. Liu, R. Jiang, M. Piao, B. Wang, H. Cao, *Clostridium butyricum*, a
butyrate-producing probiotic, inhibits intestinal tumor development through
modulating Wnt signaling and gut microbiota, *Cancer Lett* 469 (2020) 456-467.

[35] M. Liu, W. Xie, X. Wan, T. Deng, *Clostridium butyricum* modulates gut
microbiota and reduces colitis associated colon cancer in mice, *Int Immunopharmacol*
88 (2020) 106862.

[36] E.M. Velazquez, H. Nguyen, K.T. Heasley, C.H. Saechao, L.M. Gil, A.W.L.
Rogers, B.M. Miller, M.R. Rolston, C.A. Lopez, Y. Litvak, M.J. Liou, F. Faber, D.N.
Bronner, C.R. Tiffany, M.X. Byndloss, A.J. Byndloss, A.J. Bäumler, Endogenous
Enterobacteriaceae underlie variation in susceptibility to *Salmonella* infection, *Nat*
*Microbiol* 4(6) (2019) 1057-1064.

[37] S.J. Robertson, P. Lemire, H. Maughan, A. Goethel, W. Turpin, L. Bedrani, D.S.

Guttman, K. Croitoru, S.E. Girardin, D.J. Philpott, Comparison of Co-housing and
Littermate Methods for Microbiota Standardization in Mouse Models, *Cell Rep* 27(6)
(2019) 1910-1919.e2.

[38] A.C. Ericsson, J.W. Davis, W. Spollen, N. Bivens, S. Givan, C.E. Hagan, M.
McIntosh, C.L. Franklin, Effects of vendor and genetic background on the
composition of the fecal microbiota of inbred mice, *PLoS One* 10(2) (2015)
e0116704.

[39] E.F. Tigchelaar, M.J. Bonder, S.A. Jankipersadsing, J. Fu, C. Wijmenga, A.
Zhernakova, Gut microbiota composition associated with stool consistency, *Gut* 65(3)
(2016) 540-2.

[40] J. König, J. Wells, P.D. Cani, C.L. García-Ródenas, T. MacDonald, A. Mercenier,
634 J. Whyte, F. Troost, R.J. Brummer, Human Intestinal Barrier Function in Health and
635 Disease, *Clin Transl Gastroenterol* 7(10) (2016) e196.

[41] M.V. Tulstrup, E.G. Christensen, V. Carvalho, C. Linnige, S. Ahrné, O. Højberg,
637 T.R. Licht, M.I. Bahl, Antibiotic Treatment Affects Intestinal Permeability and Gut
Microbial Composition in Wistar Rats Dependent on Antibiotic Class, *PLoS One*
10(12) (2015) e0144854.

[42] A. Pellegrino, G. Coppola, F. Santopaolo, A. Gasbarrini, F.R. Ponziani, Role of
*Akkermansia* in Human Diseases: From Causation to Therapeutic Properties,
*Nutrients* 15(8) (2023).

[43] Y. Ji, Y. Yin, Z. Li, W. Zhang, Gut Microbiota-Derived Components and
Metabolites in the Progression of Non-Alcoholic Fatty Liver Disease (NAFLD),
*Nutrients* 11(8) (2019).

[44] A.K. Duttaroy, Role of Gut Microbiota and Their Metabolites on Atherosclerosis,
Hypertension and Human Blood Platelet Function: A Review, *Nutrients* 13(1) (2021).

[45] X. Jia, W. Xu, L. Zhang, X. Li, R. Wang, S. Wu, Impact of Gut Microbiota and
Microbiota-Related Metabolites on Hyperlipidemia, *Front Cell Infect Microbiol* 11
(2021) 634780.

[46] A.E. Pérez-Cobas, M.J. Gosalbes, A. Friedrichs, H. Knecht, A. Artacho, K.
Eismann, W. Otto, D. Rojo, R. Bargiela, M. von Bergen, S.C. Neulinger, C. Däumer,

F.A. Heinsen, A. Latorre, C. Barbas, J. Seifert, V.M. dos Santos, S.J. Ott, M. Ferrer, A.
Moya, Gut microbiota disturbance during antibiotic therapy: a multi-omic approach,
Gut 62(11) (2013) 1591-601.

[47] P. Louis, G.L. Hold, H.J. Flint, The gut microbiota, bacterial metabolites and
colorectal cancer, Nat Rev Microbiol 12(10) (2014) 661-72.

[48] C.M. van der Beek, C.H.C. Dejong, F.J. Troost, A.A.M. Masclee, K. Lenaerts,
Role of short-chain fatty acids in colonic inflammation, carcinogenesis, and mucosal
protection and healing, Nutr Rev 75(4) (2017) 286-305.

[49] H.J. Binder, Role of colonic short-chain fatty acid transport in diarrhea, Annu
Rev Physiol 72 (2010) 297-313.

[50] M. Wullt, M.L. Johansson Hagslätt, I. Odenholt, Lactobacillus plantarum 299v
enhances the concentrations of fecal short-chain fatty acids in patients with recurrent
clostridium difficile-associated diarrhea, Dig Dis Sci 52(9) (2007) 2082-6.

[51] G. Cresci, L.E. Nagy, V. Ganapathy, Lactobacillus GG and tributyrin
supplementation reduce antibiotic-induced intestinal injury, JPEN J Parenter Enteral
Nutr 37(6) (2013) 763-74.

[52] X. Song, X. Sun, S.F. Oh, M. Wu, Y. Zhang, W. Zheng, N. Geva-Zatorsky, R.
Jupp, D. Mathis, C. Benoist, D.L. Kasper, Microbial bile acid metabolites modulate
gut ROR γ (+) regulatory T cell homeostasis, Nature 577(7790) (2020) 410-415.

[53] W. Jia, G. Xie, W. Jia, Bile acid-microbiota crosstalk in gastrointestinal
inflammation and carcinogenesis, Nat Rev Gastroenterol Hepatol 15(2) (2018)
111-128.

[54] J.A. Winston, C.M. Theriot, Diversification of host bile acids by members of the
gut microbiota, Gut Microbes 11(2) (2020) 158-171.

**Figure legends**

**Figure 1. *C. butyricum* RH2 ameliorated antibiotic-associated diarrhea (AAD)**
**symptoms in juvenile mice.**

[revised manuscript text omitted]

734

- This study investigates the therapeutic potential of the probiotic strain *Clostridium butyricum* RH2 in alleviating antibiotic-associated diarrhea (AAD) in juvenile mice. The authors developed a murine model using a triple-antibiotic cocktail and evaluated the effects of *C. butyricum* RH2 on gut microbiota composition, intestinal barrier integrity, inflammation, and fecal metabolomics. The study includes multiple treatment groups (low, medium, high doses of RH2, antibiotic withdrawal, and controls).
- It uses well-established methods for assessing diarrhea severity, histology, gene expression, microbiota profiling, and metabolomics. Here are some suggestions to improve the article;

1. **Typographical Errors:**

- “therap eutic” → should be “therapeutic”
- “calcu lated” → should be “calculated”
- “significan ce” → should be “significance”
- “observ ed” → should be “observed”
- “chan ged” → should be “changed”

2. **Inconsistent Spacing and Hyphenation:**

- “anti - inflammatory” → should be “anti-inflammatory”
- “co - administration” → should be “co-administration”
- “microbiota - derived” → should be “microbiota-derived”

3. **Redundant or Awkward Phrasing:**

- “Given the heightened clinical vulnerability of pediatric populations to AAD complications...” → could be simplified for clarity.
- “This persistent deficit highlights the long - term detrimental effects...” → “persistent deficit” is vague; consider specifying the deficit.

4. **Subject-Verb Agreement:**

- “The microbes and metabolites could be roughly divided...” → “can be” is more appropriate in scientific writing unless referring to past analysis.

5. **Improper Use of Articles:**

- “a Gram - positive, obligate anaerobic bacterium” → correct, but sometimes “a” is missing before similar phrases.

Technical Mistakes and Clarity Issues

1. Ambiguous Dose Descriptions:

- “low (LD: 1.05×10^5 CFU daily)” → consider clarifying whether this is per mouse or per group.

2. Inconsistent Terminology:

- “fecal water content” and “water in stool” are used interchangeably; standardize terminology.

3. Missing Units or Definitions:

- “Figure 1B” mentions body weight but doesn’t specify units (grams?).

4. Overuse of Passive Voice:

- “Samples were immediately weighed...” → consider active voice for clarity: “We immediately weighed the samples...”

5. Scientific Jargon Without Explanation:

- Terms like “LEfSe analysis,” “Faith’s PD,” and “SMPDB” are used without brief definitions, which may confuse non-specialist readers.

I will recommend accepting the article after the above mentioned changes.

Dr.SADIA ALAM

Associate Professor Microbiology

The University of Haripur, Pakistan

Reviewer #1 (Comments for the Author):

From my perspective of view, in this manuscript:

Line: 36 Repetition: the phrases "young mice" and "juvenile mice" appeared twice; the term can be unified.

Response: Thank you for the comment. To maintain consistency in terminology, we have now unified the expression and revised all instances of "young mice" to "juvenile mice" throughout the manuscript.

Line 45: The flow between the sentence describing the results and the concluding sentence can be improved (the last sentence appears more appended than integrated).

Response: Thank you for the helpful comment. We have revised the abstract to improve the logical flow between the results and the conclusion. Specifically, we have rephrased the final sentence for smoother integration as follows: "Collectively, these findings indicate that the therapeutic benefits of *C. butyricum* are closely linked to its ability to tolerate continuous antibiotic exposure, providing a scientific rationale for its co-administration with antibiotics."

Line 151 : The reference (housekeeping) gene used for normalizing gene expression (such as GAPDH or β -actin) was not specified.

Response: Thank you for the comment. We apologize for the oversight of not specifying the internal reference gene in the original manuscript. As detailed in Supplementary Table 1, we used GAPDH as the housekeeping gene. This information has now been added to the "RNA extraction and RT-PCR" section of the manuscript as follows: "Gene expression levels were normalized to GAPDH and calculated using the $2^{(-\Delta \Delta Ct)}$ method, as previously described."

Line 153: The reaction volume, RNA concentration, and thermal cycling conditions for qPCR were not specified, and these are often required.

Response: Thank you for the comment. We apologize for not specifying the detailed information of qPCR method in the original manuscript. These details have now been added to the "RNA extraction and RT-PCR" section as "Quantitative real-time PCR (qPCR) was performed in a 10 μ L

reaction volume containing 1 μL of cDNA template, 5 μL of ChamQ Universal SYBR qPCR Master Mix, 0.4 μL of each primer (10 μM), and 3.2 μL of nuclease-free water. The reactions were run under the following thermal cycling conditions: initial denaturation at 95°C for 30 seconds, followed by 40 cycles of 95°C for 10 seconds and 60°C for 30 seconds.”.

Line 221: For better flow: the transition from describing the experiment to presenting the results (weight loss, lethargy) was abrupt; it would be preferable to add a linking sentence.

Response: We thank the reviewer for this helpful suggestion. To address this, we have now added a linking sentence to improve the flow between the experimental description and the results as follows: “After establishing the AAD model and assigning mice into the respective groups, we next evaluated the general health status of the animals. AAD juvenile mice exhibited significant weight loss compared to healthy controls, accompanied by clinical manifestations including lethargy and reduced activity.”

Line 369: Given that SCFAs and bile acids play central roles in gut homeostasis, how can we determine whether the observed therapeutic effect of *C. butyricum* RH2 is primarily driven by SCFA production, bile acid modulation, or synergistic interactions between both pathways?

Response: We thank the reviewer for raising this important mechanistic question. We agree that our current data only demonstrate correlations between metabolites and clinical outcomes, and therefore cannot establish causality. While our analysis showed that SCFAs (particularly butyrate) displayed stronger and more consistent associations with diarrhea relief, reduced inflammation, and improved barrier function (Figure 6), we cannot exclude the possibility that bile acid modulation also contributes, either independently or through synergistic interactions.

Accordingly, we have revised the Discussion to emphasize that the therapeutic effect of *C. butyricum* RH2 may arise from a combination of SCFA restoration and bile acid modulation, but the relative contributions of these pathways remain to be clarified. To address this, future studies using receptor knockout models (e.g., GPR109A for SCFAs, FXR for bile acids) or pathway-specific inhibitors are planned to dissect the causal mechanisms.

Line 434: Given that *C. butyricum* is a transient rather than colonizing species, to what extent can

its long-term ecological and therapeutic effects be sustained after discontinuation of supplementation, and how might this impact its clinical applicability in AAD treatment?

Response: We thank the reviewer for this important question. We agree that *C. butyricum* RH2 is considered a transient species and that our current study did not include a post-supplementation washout phase; therefore, we cannot determine from these data whether ecological or therapeutic effects persist after discontinuation. In our model, RH2 was administered under continued antibiotic exposure and compared with an antibiotic-withdrawal control, but no follow-up after stopping RH2 was performed.

Within these limits, our results suggest two plausible routes by which benefits might outlast the physical presence of RH2: (i) RH2 was associated with increases in autochthonous taxa linked to barrier function and stool firmness (e.g., *Akkermansia*, *Ruminococcus*, *Oscillospira*, *Bacteroides*) and reductions in pathobionts (e.g., *Acinetobacter*, *Staphylococcus*) ; and (ii) concomitant improvements in barrier-related transcripts (e.g., *Ocln*, *Tjp1*, *Muc2*, *Cdh1*, *Cldn1/5*) were observed . These observations are consistent with a mechanism where RH2 may “catalyze” recovery of the endogenous ecosystem and barrier function; however, they remain correlative and do not establish durability or causality.

Regarding clinical applicability to AAD, sustained colonization may not be necessary if RH2 primarily accelerates microbiome/ barrier recovery during and immediately after antibiotic therapy. Practically, this would support use during the antibiotic course with a short taper post-antibiotic, while recognizing that durability needs prospective verification.

To address the reviewer’s point, we have revised the Discussion to explicitly acknowledge this limitation and outline future work (e.g., adding a washout arm with serial sampling after RH2 cessation, strain-tracking/qPCR for RH2, time-series metagenomics and SCFA profiling, and perturbation-resilience assays) to test persistence and causality.

Furthermore, we include a paragraph describing the limitations of this study and possible strategies to address them in future studies at the end of the discussion section as follows: " Despite the promising findings, this study has limitations that warrant consideration. Because *C. butyricum* RH2 is a transient species and our study did not include a post-supplementation washout phase, we cannot determine whether ecological or clinical benefits persist after discontinuation. Although RH2 was associated with increased abundances of beneficial

autochthonous taxa and improved barrier-related transcripts under continuous antibiotic exposure, these findings are correlative. Future studies incorporating a washout arm, strain-specific tracking, and time-series multi-omics are needed to establish durability and causality."

Reviewer #2 (Comments for the Author):

- This study investigates the therapeutic potential of the probiotic strain *Clostridium butyricum* RH2 in alleviating antibiotic-associated diarrhea (AAD) in juvenile mice. The authors developed a murine model using a triple-antibiotic cocktail and evaluated the effects of *C. butyricum* RH2 on gut microbiota composition, intestinal barrier integrity, inflammation, and fecal metabolomics. The study includes multiple treatment groups (low, medium, high doses of RH2, antibiotic withdrawal, and controls).
- It uses well-established methods for assessing diarrhea severity, histology, gene expression, microbiota profiling, and metabolomics. Here are some suggestions to improve the article;

1. Typographical Errors:

- "therap eutic" → should be "therapeutic"
- "calcu lated" → should be "calculated"
- "significan ce" → should be "significance"
- "observ ed" → should be "observed"
- "chan ged" → should be "changed"

Response: We thank the reviewer for carefully identifying these typographical errors. We apologize for the oversight. All the listed mistakes (“therap eutic,” “calcu lated,” “significan ce,” “observ ed,” and “chan ged”) have been corrected to “therapeutic,” “calculated,” “significance,” “observed,” and “changed,” respectively. In addition, we have conducted a thorough review of the entire manuscript to ensure that similar typographical errors have been corrected throughout.

2. Inconsistent Spacing and Hyphenation:

- "anti - inflammatory" → should be "anti-inflammatory"
- "co - administration" → should be "co-administration"
- "microbiota - derived" → should be "microbiota-derived"

Response: We thank the reviewer for pointing out these inconsistencies in spacing and hyphenation. We apologize for the oversight. The terms have been corrected as suggested

(“anti-inflammatory,” “co-administration,” and “microbiota-derived”), and we have carefully reviewed the entire manuscript to ensure that compound words and prefixes are now used consistently.

3. Redundant or Awkward Phrasing:

- "Given the heightened clinical vulnerability of pediatric populations to AAD complications..." → could be simplified for clarity.
- "This persistent deficit highlights the long - term detrimental effects..." → "persistent deficit" is vague; consider specifying the deficit.

Response: Thank you for your helpful comments. The first sentence has been simplified for clarity as follows: “Given that pediatric populations are particularly vulnerable to AAD complications ...”. For the second sentence, the vague phrase “persistent deficit” has been replaced with a more specific description: “This incomplete recovery of barrier-related gene expression highlights the long-term detrimental effects of antibiotic exposure.”

4. Subject-Verb Agreement:

- "The microbes and metabolites could be roughly divided..." → "can be" is more appropriate in scientific writing unless referring to past analysis.

Response: Thank you for the comment. We have revised the sentence to correct the subject–verb agreement. It now reads: “The microbes and metabolites can be roughly divided ...”

5. Improper Use of Articles:

- "a Gram - positive, obligate anaerobic bacterium" → correct, but sometimes "a" is missing before similar phrases.

Response: Thank you for the suggestion. I revised the sentence to conform to standard grammatical conventions. Thank you for the comment. We have carefully reviewed the manuscript and corrected the improper use of articles. For example, phrases such as “Gram-positive, obligate anaerobic bacterium” have now been consistently written as “a Gram-positive, obligate anaerobic bacterium.”

Technical Mistakes and Clarity Issues

1. Ambiguous Dose Descriptions:

- "low (LD: 1.05×10^5 CFU daily)" → consider clarifying whether this is per mouse or per group.

Response: Thank you for the comment. The values have been now reported as "CFU daily per mice" to more accurately reflect the dosing regimen.

2. Inconsistent Terminology:

- "fecal water content" and "water in stool" are used interchangeably; standardize terminology.

Response: Thank you for the suggestion. I have uniformly used the term "water in stool" has now been unified throughout the manuscript for consistency.

3. Missing Units or Definitions:

- "Figure 1B" mentions body weight but doesn't specify units (grams?).

Response: Thank you for the comment. We've added the weight unit (g) to the vertical axis of Figure 1B.

4. Overuse of Passive Voice:

- "Samples were immediately weighed..." → consider active voice for clarity: "We immediately weighed the samples..."

Response: Thank you for the comment. The sentence has been revised according to your suggestion.

5. Scientific Jargon Without Explanation:

- Terms like "LEfSe analysis," "Faith's PD," and "SMPDB" are used without brief definitions, which may confuse non-specialist readers.

Response: Thank you for the comment. We have now provided concise definitions and explanations for the specialized terms upon their first appearance in the text.

Reviewer #3 (Public repository details (Required)):

16S rRNA gene sequence.

Response: Sequencing datasets have been deposited to the NCBI Sequence Read Archive under BioProject accession numbers PRJNA1157009.

Reviewer #3 (Comments for the Author):

Dear Authors,

Thanks for your work.

Line 75: It is important to highlight *Clostridium* spp is not yet considered GRAS for probiotic applications unlike *Bifidobacterium* and *Lactobacillus* that have been widely studied and accepted as GRAS.

Response: We thank the reviewer for this important comment. We agree that it is necessary to clearly highlight the GRAS status differences. Accordingly, we have revised the Introduction to explicitly state that *Clostridium* spp. is not yet considered GRAS for probiotic applications, in contrast to *Bifidobacterium* and *Lactobacillus* which have been widely accepted as GRAS. We also noted that while *Clostridium butyricum* has not been formally granted GRAS status by the US FDA, the safety of specific strains such as MIYAIRI 588 has been well documented through long-term clinical use. Moreover, our study employed the RH2 strain, which has been approved and marketed in China (as *Clostridium butyricum* Live Bacterial Capsules, Misang®) since 2004, with a strong safety record and no serious adverse events reported.

At the end of the discussion section, please include a paragraph describing the limitations of this study and possible strategies to address them in future studies.

Response: Thank you for the comment. We have now incorporated a paragraph at the end of the Discussion section to outline the limitations of this study and to suggest potential strategies for addressing them in future research as follows: " Despite the promising findings, this study has limitations that warrant consideration. Because *C. butyricum* RH2 is a transient species and our study did not include a post-supplementation washout phase, we cannot determine whether

ecological or clinical benefits persist after discontinuation. Although RH2 was associated with increased abundances of beneficial autochthonous taxa and improved barrier-related transcripts under continuous antibiotic exposure, these findings are correlative. Future studies incorporating a washout arm, strain-specific tracking, and time-series multi-omics are needed to establish durability and causality."

Re: Spectrum01976-25R1 (***Clostridium butyricum* RH2 ameliorates Diarrhea in juvenile mice under continuous antibiotics exposure by modulating gut microbiota and metabolome**)

Dear Dr. Yu Wang:

Thank you for the privilege of reviewing your work. Below you will find my comments, instructions from the Spectrum editorial office, and the reviewer comments.

Revision Guidelines

Sincerely,
Yuan Pin Hung
Editor
Microbiology Spectrum

Reviewer #3 (Comments for the Author):

Dear Authors,

Thank you for addressing all the concerns raised in the initial review; no further concerns remain.

Reviewer #4 (Comments for the Author):

AAD remains an important clinical challenge, especially in pediatric populations and this work explores therapeutic possibilities using a juvenile mouse model under continuous antibiotic exposure. The therapeutic efficacy of *Clostridium butyricum* RH2 strain in managing AAD has not been characterized in the past. However, a stronger mechanistic connection will be beneficial, considering that in almost all the experiments, the differences between AW and HD groups are modest at the best.

Major comments:

1. The finding that RH2 can maintain its therapeutic effects even under ongoing antibiotic treatment is promising. However, all the data presented are correlative associations between RH2 supplementation and microbiota changes, improved barrier etc. In my opinion, at least one mechanistic link will make the manuscript stronger. Have the authors tried supplementing metabolites (e.g. SCFA) to recapitulate the observations. In the absence of any link, it's difficult to conclude that RH2 is causing these effects.
2. Consider expanding the relevant discussion to interpret the metabolomic results within known host- microbiota metabolic pathways relevant to AAD.
3. Fig. 1-C and 1-D: Are the differences between AW and HD group biologically relevant?

Minor comments:

1. Discussion in general is too lengthy without actually connecting any of the results to published mechanisms relevant to AAD.
2. Fig. 4-C: AW and HD look similar, but LD or MD are very different from HD, explain.

AAD remains an important clinical challenge, especially in pediatric populations and this work explores therapeutic possibilities using a juvenile mouse model under continuous antibiotic exposure. The therapeutic efficacy of *Clostridium butyricum* RH2 strain in managing AAD has not been characterized in the past. However, a stronger mechanistic connection will be beneficial, considering that in almost all the experiments, the differences between AW and HD groups are modest at the best.

Major comments:

1. The finding that RH2 can maintain its therapeutic effects even under ongoing antibiotic treatment is promising. However, all the data presented are correlative associations between RH2 supplementation and microbiota changes, improved barrier etc. In my opinion, at least one mechanistic link will make the manuscript stronger. Have the authors tried supplementing metabolites (e.g. SCFA) to recapitulate the observations. In the absence of any link, it's difficult to conclude that RH2 is causing these effects.
2. Consider expanding the relevant discussion to interpret the metabolomic results within known host-microbiota metabolic pathways relevant to AAD.
3. Fig. 1-C and 1-D: Are the differences between AW and HD group biologically relevant?

Minor comments:

1. Discussion in general is too lengthy without actually connecting any of the results to published mechanisms relevant to AAD.
2. Fig. 4-C: AW and HD look similar, but LD or MD are very different from HD, explain.

Reviewer #4 (Comments for the Author):

AAD remains an important clinical challenge, especially in pediatric populations and this work explores therapeutic possibilities using a juvenile mouse model under continuous antibiotic exposure. The therapeutic efficacy of *Clostridium butyricum* RH2 strain in managing AAD has not been characterized in the past. However, a stronger mechanistic connection will be beneficial, considering that in almost all the experiments, the differences between AW and HD groups are modest at the best.

AU: We thank the reviewer for the thoughtful overall assessment and constructive feedback. We appreciate the comments regarding the clinical relevance, the modest differences between AW and HD groups, and the need for stronger mechanistic interpretation. These points are well taken, and we have addressed them in detail in the responses to the major comments below.

Major comments:

1. The finding that RH2 can maintain its therapeutic effects even under ongoing antibiotic treatment is promising. However, all the data presented are correlative associations between RH2 supplementation and microbiota changes, improved barrier etc. In my opinion, at least one mechanistic link will make the manuscript stronger. Have the authors tried supplementing metabolites (e.g. SCFA) to recapitulate the observations. In the absence of any link, it's difficult to conclude that RH2 is causing these effects.

AU: We sincerely appreciate the reviewer's insightful suggestion. Indeed, we fully agree that establishing a more explicit mechanistic link would further strengthen the manuscript. At present, our data primarily demonstrated associations between RH2 supplementation, increased abundance of *Clostridium butyricum*, and shifted in the gut microbiota composition, and improved epithelial barrier function. As these findings remain correlative, we have now revised the conclusion to avoid implying a definitive causal relationship.

The reviewer's suggestion to examine whether specific metabolites—particularly SCFAs—can recapitulate the observed effects is highly valuable. SCFAs, especially butyrate, are well recognized for their roles in supporting barrier integrity and modulating mucosal inflammation. In this study, although we did not perform metabolite supplementation experiments, we did observe that elevated SCFA levels (notably butyrate) were consistently associated with

improved expression of barrier-related genes (such as *Ocln*, *Cldn1*, and *Muc2*) and reduced diarrhea severity. These observations point to a plausible SCFA-mediated pathway underlying RH2's beneficial effects. Accordingly, we have revised the corresponding sentence in the Discussion section (Lines 436-443) to clarify the potential mechanism.

In addition, we have further expanded the Discussion (Lines 436-443) to articulate how SCFA-related signaling may contribute to RH2-induced improvements and to outline future studies, such as butyrate supplementation or receptor-blocking approaches—that would allow us to directly test this hypothesis. We hope these revisions better contextualize the current correlative implications and provide a clearer framework for future mechanistic investigations.

2. Consider expanding the relevant discussion to interpret the metabolomic results within known host-microbiota metabolic pathways relevant to AAD.

AU: We appreciate this valuable suggestion. Accordingly, in the revised manuscript, we have now expanded the Discussion to better interpret the metabolomic findings within known host-microbiota metabolic pathways implicated in AAD. Specifically, we have now discussed how alterations in SCFAs, bile acid derivatives, and linoleic / α -linolenic acid metabolism observed in our study (Figure 5) relate to intestinal inflammation, epithelial repair, and PPAR- γ -associated anti-inflammatory lipid mediators (Yuan, Chen et al. 2015). We also have now highlighted that the partial restoration of the urea cycle intermediates and bile acid derivatives (e.g., taurodeoxycholic acid) reflects normalization of microbial amino acid and bile acid transformation pathways commonly disrupted in AAD. These revisions strengthen the mechanistic relevance of the metabolomic data in the manuscript.

3. Fig. 1-C and 1-D: Are the differences between AW and HD group biologically relevant?

AU: Thank you for raising this important point. We agree that the differences between the AW and HD groups are modest. However, they remain statistically significant and more importantly, reflect two biologically distinct conditions. Mice in the AW group experience spontaneous recovery once antibiotics are withdrawn, whereas HD mice continue to receive antibiotics yet still show significant improvement following *C. butyricum* RH2 supplementation. This indicates that *C.*

butyricum RH2 can exert therapeutic benefits even under ongoing antibiotic pressure—a clinically relevant scenario in pediatric AAD where discontinuation of antibiotics is often not feasible.

Minor comments:

1. Discussion in general is too lengthy without actually connecting any of the results to published mechanisms relevant to AAD.

AU: We appreciate the reviewer's feedback. We have substantially streamlined the Discussion and reorganized it into a more mechanistic and hypothesis-driven narrative. Redundant descriptions have been removed, and the revised Discussion section now more clearly integrates our findings with published mechanisms relevant to AAD, including epithelial barrier dysfunction, SCFA signaling, bile acid metabolism, and gut microbiota-driven immune regulation.

2. Fig. 4-C: AW and HD look similar, but LD or MD are very different from HD, explain.

AU: We thank the reviewer for this comment. We interpret these findings as reflecting a dose-dependent effect of *Clostridium butyricum* RH2. At the low and medium doses (LD/MD), the administered amount of RH2 seems insufficient to restore key microbial taxa or induce substantial metabolic shifts. In contrast, the high dose (HD) appears to reach an effective threshold that enables more pronounced community-level changes and supports metabolite recovery, resulting in a microbiota profile more similar to that of the AW/NC groups. This pattern is consistent with dose-dependent responses reported for other probiotics in clinical studies (Johnston, Supina et al. 2006, Ouwehand 2017).

References

Johnston, B. C., A. L. Supina and S. Vohra (2006). "Probiotics for pediatric antibiotic-associated diarrhea: a meta-analysis of randomized placebo-controlled trials." Cmaj **175**(4): 377-383.

Ouwehand, A. C. (2017). "A review of dose-responses of probiotics in human studies." Benef Microbes **8**(2): 143-151.

Yuan, G., X. Chen and D. Li (2015). "Modulation of peroxisome proliferator-activated receptor gamma (PPAR γ) by conjugated fatty acid in obesity and inflammatory bowel disease." Journal of agricultural and food chemistry **63**(7): 1883-1895.

Re: Spectrum01976-25R2 (*Clostridium butyricum* RH2 ameliorates Diarrhea in juvenile mice under continuous antibiotics exposure by modulating gut microbiota and metabolome)

Dear Dr. Yu Wang:

Your manuscript has been accepted, and I am forwarding it to the ASM production staff for publication. Your paper will first be checked to make sure all elements meet the technical requirements. ASM staff will contact you if anything needs to be revised before copyediting and production can begin. Otherwise, you will be notified when your proofs are ready to be viewed.

Sincerely,
Yuan Pin Hung
Editor
Microbiology Spectrum

Reviewer #3 (Comments for the Author):

Dear authors,

Thanks for addressing all the review concerns raised.